# Continual Optimization with Symmetry Teleportation for Multi-Task Learning

**Zhipeng Zhou**[1], **Ziqiao Meng**[2], **Pengcheng Wu**[1], **Peilin Zhao**[3], **Chunyan Miao**[1*]

[1]Nanyang Technological University
[2]National University of Singapore
[3]School of Artificial Intelligence, Shanghai Jiao Tong University
zzpustcml@gmail.com, zq-meng@nus.edu.sg, peilinzhao@sjtu.edu.cn,
{pengchengwu, ascymiao}@ntu.edu.sg

## Abstract

Multi-task learning (MTL) is a widely explored paradigm that enables the simultaneous learning of multiple tasks using a single model. Despite numerous solutions, the key issues of optimization conflict and task imbalance remain under-addressed, limiting performance. Unlike existing optimization-based approaches that typically reweight task losses or gradients to mitigate conflicts or promote progress, we propose a novel approach based on **C**ontinual **O**ptimization with **S**ymmetry **T**eleportation (COST). During MTL optimization, when an optimization conflict arises, we seek an alternative loss-equivalent point on the loss landscape to reduce conflict. Specifically, we utilize a low-rank adapter (LoRA) to facilitate this practical teleportation by designing convergent, loss-invariant objectives. Additionally, we introduce a historical trajectory reuse strategy to continually leverage the benefits of advanced optimizers. Extensive experiments on multiple mainstream datasets demonstrate the effectiveness of our approach. COST is a plug-and-play solution that enhances a wide range of existing MTL methods. When integrated with state-of-the-art methods, COST achieves superior performance. Code is avaliable at https://github.com/zzpustc/COST.

## 1 Introduction

Traditional machine learning typically requires separate models for each task, leading to higher computational and storage demands as the number of tasks increases[Zhou et al., 2023, Pan et al., 2024, Feng et al., 2024]. To overcome this issue, multi-task learning (MTL) offers an efficient approach, enabling the simultaneous learning of multiple tasks using a single model [Qiu et al., 2017, Hai et al., 2016, Yao et al., 2021]. Recent developments in

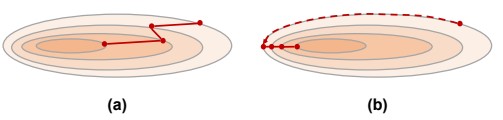

Figure 1: The illustration of symmetry teleportation. (a) is the original gradient descent. (b) is the gradient descent with a faster convergence rate after teleporting the start point from (a).

MTL methods can be broadly divided into two categories: structure-based [Heuer et al., 2021, Ye and Xu, 2023, Chen et al., 2023] and optimization-based [Sener and Koltun, 2018, Yu et al., 2020, Liu et al., 2024a]. Structure-based methods focus on designing architectures that enhance task learning by utilizing task relationships and promoting individual progress. On the other hand, optimization-based methods prioritize the learning process by addressing challenges such as gradient conflicts and task imbalances. Since this paper concentrates on optimization-based methods, our analysis and comparisons will primarily focus on these approaches.

---

*Corresponding author.

Optimization-based MTL aims to resolve the aforementioned issues by re-weighting on various aspects. For example, a series of studies [Sener and Koltun, 2018, Yu et al., 2020, Liu et al., 2021a] explore different gradient combinations to prevent improving some tasks while sacrificing others. Another group of works [Chen et al., 2018, Liu et al., 2024a] re-weight task loss to ensure fair progress for individual tasks and thereby address the imbalance issue. While the former group of works endeavors to balance conflict and imbalance issues, the latter focuses on attaining balanced individual progress with minimal concern for the conflict issue. As a result, the latter is generally less robust in different scenarios compared to the former according to empirical observations [Chen et al., 2018]. However, the former also struggles to achieve a proper balance. In this paper, distinct from these two paradigms and based on the definition of Pareto dominance, we approach MTL from a new perspective, i.e., seeking the less conflicting and more convergent point through symmetry teleportation during MTL optimization.

Unlike the traditional gradient descent regime, symmetry teleportation aims to accelerate the optimization process by seeking another point within the same loss level set, as depicted in Figure 1. Several recent works [Armenta et al., 2023, Zhao et al., 2022, 2023] have explored optimization through symmetry teleportation. For example, [Zhao et al., 2022] introduces a simple teleportation algorithm for non-linear neural networks, based on the assumption that activation functions are bijective, and seeks the point of maximal gradient magnitude using gradient ascent. However, these methods do not provide practical algorithms for larger, modern neural networks, primarily due to their reliance on strict assumptions about non-linearity and computational intensity (Section 3.3). These limitations are especially pronounced in more complex tasks, e.g., MTL.

Therefore, in this paper, we aim to develop a practical symmetry teleportation method that is applicable for modern deep models, and addressing MTL issues. Specifically, we leverage the low-rank adapter (LoRA) to realize teleportation when encountered with the conflicts issue. By designing the objectives to ensure the invariant task loss and promote progress, we are able to further extend the boundaries of individual task learning for MTL models in a balanced manner. Besides, we also design a historical trajectory reuse strategy to continually benefit from advanced optimizer (e.g., Adam). In a nutshell, our contribution can be summarized as follows:

- We approach MTL from a new angle, i.e., symmetry teleportation, and empirically verify its applicability for MTL (Section 3.2).

- A new practical teleportation method COST is proposed for mitigating the conflict and imbalance issue. To the best of our knowledge, we are the first to develop a practical teleportation method for non-small deep models, specifically for MTL.

- By proposing a historical trajectory reuse strategy, we can continually benefit from the advanced optimizer (e.g., Adam and its variants).

- Taking the advanced method as the baseline, our COST can well augment it to achieve state-of-the-art (SOTA) performance across diverse evaluations. Besides, we also equip mainstream MTL methods with COST, and showing its plug-and-play property.

## 2 Related Work

### 2.1 Optimization-based MTL

Optimization-based methods aim to optimize multiple tasks simultaneously by enhancing the gradient-based learning process itself. For example, MGDA [Sener and Koltun, 2018] reduces conflicts between task gradients by combining them using the Frank-Wolfe algorithm [Jaggi, 2013] to generate a gradient with minimal norm. PCGrad [Yu et al., 2020] addresses gradient conflicts by projecting gradients from different tasks onto directions that minimize interference. CAGrad [Liu et al., 2021a] attempts to balance global optimization and task-specific performance, maintaining both Pareto efficiency and overall optimization with the assistance of a hyperparameter. Nash-MTL [Navon et al., 2022] introduces a game-theoretic approach where tasks negotiate to update parameters in a manner that enables balanced progression across tasks. Additionally, MoCo [Fernando et al., 2023] focuses on correcting biases in gradient direction by tracking parameters during the learning process, improving gradient alignment and task performance. FairGrad [Ban and Ji, 2024] is a pioneering MTL algorithm that puts forward fairness measurements to facilitate maximal loss reduction. It can

be considered as an advanced version of Nash-MTL, being capable of balancing task progress in a more fine-grained manner.

## 2.2 Symmetry Teleportation for Deep Model

Before presenting some recent works on symmetry teleportation, we first provide its definition here as per [Zhao et al., 2022]. Let $\mathcal{L}(\boldsymbol{\theta})$ be the loss function. Here, $\mathbb{R}^d$ denotes the model's parameter space, and $A$ represents the acting space on the parameters that leaves the loss value unchanged. Subsequently, we have the following definition:

$$\mathcal{L}(\boldsymbol{\theta}) = \mathcal{L}(a \cdot \boldsymbol{\theta}), \quad \forall a \in A, \quad \forall \boldsymbol{\theta} \in \mathbb{R}^d. \tag{1}$$

$$\boldsymbol{\theta}' = a \cdot \boldsymbol{\theta}, \quad a = \underset{a \in A}{\operatorname{argmax}} \left\| \nabla \mathcal{L}(a \cdot \boldsymbol{\theta}) \right\|^2. \tag{2}$$

we can observe that symmetry teleportation aims to find a loss-invariant point (Eqn. 1) with a maximum gradient norm (Eqn. 2) on the loss level set by acting with a group element.

As a recent research topic, symmetry teleportation has been explored in only a few works [Armenta et al., 2023, Zhao et al., 2022, 2023]. [Armenta et al., 2023] first introduced the concept of 'neural teleportation' and investigated its impact on optimization. [Zhao et al., 2022] proposed a gradient ascent-based teleportation algorithm for small neural networks (e.g., three-layer MLPs). And [Zhao et al., 2023] established the connection between symmetry teleportation and generalization through a series of theoretical analyses and provided an alternative for enhancing the meta optimizer.

## 2.3 Low-Rank Adapter

LoRA is gaining increasing popularity in tandem with the rapid advancement of foundation models and parameter-efficient fine-tuning (PEFT). It operates by maintaining the pre-trained weights of a large model in a fixed state and incorporating small, trainable rank decomposition matrices. During fine-tuning, rather than modifying all the parameters of the model, only these low-rank matrices are subject to update.

Moreover, LoRA has several variants that can attain dynamic rank [Zhang et al.], or quantization [Dettmers et al., 2024]. For instance, AdaLoRA [Zhang et al.] adaptively assigns dynamic rank to different parameters, thereby enabling the capture of important updates while preserving efficiency. In contrast, QLoRA [Dettmers et al., 2024] introduces 4-bit NormalFloat, double quantization, and paged optimizers to more effectively optimize LoRA, while significantly reducing the required memory.

**Connection and Difference**: Our work tackles conflict and imbalance issues in optimization-based MTL through symmetry teleportation. Specifically, we utilize LoRA to implement practical teleportation. In contrast to previous studies, we explore MTL from a novel perspective and introduce a new teleportation algorithm for modern deep models. This algorithm is scalable, easily integratable, and compatible with both PEFT and MTL.

# 3 Motivation and Observation

## 3.1 Preliminary

As mentioned, optimization-based MTL approaches operate under the assumption that the model consists of a task-shared backbone network alongside task-specific branches. Consequently, the primary objective of these approaches is to devise gradient combination strategies that optimize the backbone network to yield benefits across all tasks. Let us consider a scenario where there are $K \geq 2$ tasks available, each associated with a differentiable loss function $\mathcal{L}_i(\boldsymbol{\theta})$, where $\boldsymbol{\theta}$ represents the task-shared

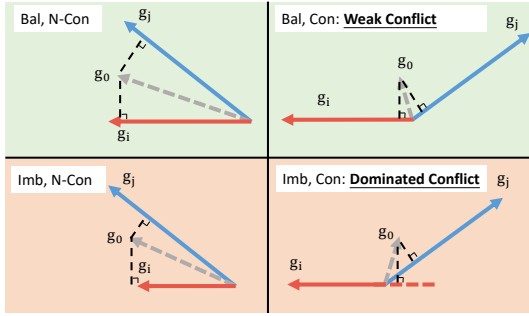

Figure 2: Illustration of conflict and imbalance issues in MTL. 'Bal' and 'Imb' represent balanced and imbalanced, while 'N-Con' and 'Con' represent non-conflicting and conflicting.

parameters. The goal of optimization-based MTL is to search for the optimal $\boldsymbol{\theta}^* \in \mathbb{R}^m$ that minimizes the losses for all tasks.

**Definition 1** (**Gradient Similarity**). Denote $\phi_{ij}$ as the angle between two task gradients $\boldsymbol{g_i}$ and $\boldsymbol{g_j}$, and assume $\|\boldsymbol{g_i}\|_2 \leq \|\boldsymbol{g_j}\|_2$, then we define the gradient similarity as $\cos \phi_{ij}$ and the gradients as conflicting when $\cos \phi_{ij} < 0$ (referred as **Weak Conflict**). When the mean gradient $\boldsymbol{g_0}$ is conflicting with $\boldsymbol{g_i}$, we call it as **Dominated Conflict** (see Figure 2).

## 3.2 Applicability of Symmetry Teleportation

Before delving into the principal design of our method, it is necessary to verify the existence of parameter symmetries with differing conflict statuses. To this end, we examine the optimization process of mainstream MTL approaches. We analyze the mean loss across all tasks and the associated conflict status during optimization from various initial points, with the results presented in Figure 5.

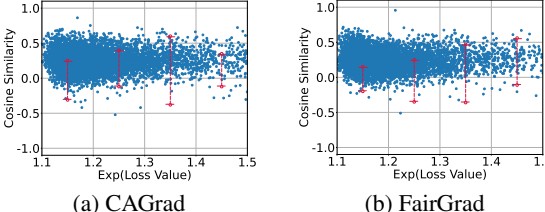

(a) CAGrad      (b) FairGrad

As shown in Figure 5, it is often possible to identify a non-conflict alternative at the same loss level when encountering conflict, demonstrating the potential of symmetry teleportation. Additional statistical results from other MTL approaches are provided in the **Appendix** (Sec. A.3).

Figure 3: Dominated conflict vs. loss examination. The pink backdrop designates the conflicting area, whereas the green backdrop indicates the nonconflicting area. The blue scatter points are the individual recorded points throughout the optimization process. The red dashed line symbolizes the teleportation occurring from a conflict point to a nonconflict point. An exponential amplification has been applied to the loss values to enhance visual clarity.

## 3.3 Pitfall of Current Paradigms

While several works [Armenta et al., 2023, Zhao et al., 2022, 2023] have proposed symmetry teleportation algorithms for neural network-based models, we demonstrate their limitations with current deep models. First, these algorithms require activation functions to be bijective, which poses a significant challenge for widely-used deep models (e.g., ResNet-50) that use non-bijective activation functions, e.g., ReLU and Sigmoid. Second, they require calculating the pseudo-inverse of inputs layer by layer to ensure output and loss invariance. This process is computationally intensive and may be impractical for modern deep models. As a result, these approaches have only been tested on simple three-layer MLP networks and small-scale datasets (e.g. MNIST) for verification.

# 4 Principal Design

In this section, we present the detailed design of COST, incorporating the symmetry teleportation paradigm and a historical trajectory reuse strategy. We also provide an analysis of convergence.

## 4.1 Continual Optimization with Symmetry Teleportation

The overall framework of COST is depicted in Figure 4. At a certain training stage $t$, we utilize LoRA to teleport the weight of the shared backbone to the non-conflict point (merge the trained LoRA into the backbone's weight) with the same loss level. Subsequently, the model (including both the backbone and branches) is continuously optimized by other MTL algorithms. In this framework, there are two questions that need to be answered:

**When**: The first question is, when should teleportation be triggered? Unfortunately, the previous solutions presented in [Armenta et al., 2023, Zhao et al., 2022, 2023] did not offer a clear answer to this question. They merely triggered it in a random or intuitive manner. In contrast, our goal is to address two key challenges in MTL: conflict and imbalance, challenges that are not concurrently addressed by existing solutions [Zhou et al., 2025, 2024]. Moreover, a naïve linear scalarization (LS) strategy can effectively promote all tasks, as illustrated in Figure 2 and has been empirically verified in [Xin et al., 2022]. Thus, the primary challenge lies in resolving conflict arising from imbalance, i.e., dominated conflict. Therefore, we establish the teleportation trigger condition based on the

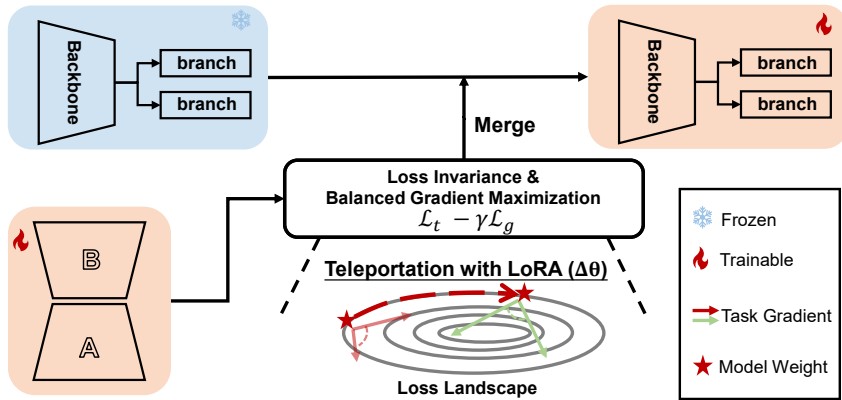

Figure 4: The Illustration of COST. Here, we depict a one-time teleportation procedure by using a 2-task example for the sake of illustration. **Note that LoRA is only applied to the shared backbone**.

occurrence of dominated conflict [2]:

$$\cos \phi_{i0} < 0, \quad \phi_{i0} = \angle(\boldsymbol{g_i}, \boldsymbol{g_0}) \tag{3}$$

where $\boldsymbol{g_i}$ and $\boldsymbol{g_0}$ represent the task gradient with the smallest norm and the mean gradient, respectively. However, when handling a large number of tasks, dominated conflicts become inevitable, reaching a 97% conflict ratio per epoch on CelebA [Liu et al., 2015], as shown in Figure 5(a). Then if we still employ dominated conflict as the trigger condition, frequent teleportation would occurs and results in inefficiency. Therefore, our objective shifts to mitigating dominant conflicts, balancing efficiency and effectiveness. To achieve this, we adopt the following condition:

$$\sum_{j}^{K} \mathbb{1}[\cos \phi_{j0} < 0] \geq \left\lceil \frac{K}{2} \right\rceil \tag{4}$$

Under this condition, the trigger frequency is significantly reduced (see Figure 5(a)) while maintaining effectiveness, as demonstrated in the evaluation. Additionally, we analyze the trade-off between effectiveness and efficiency for this condition in the **Appendix** (Section A.4).

**How**: In the symmetry teleportation paradigm, there are two key objectives: loss invariance and gradient maximization, as outlined in Eqn. 1 and Eqn. 2. Since finding a group action $g$ is infeasible for deep models, we instead use LoRA ($\boldsymbol{\Delta\theta}$) as an alternative, reformulating it as:

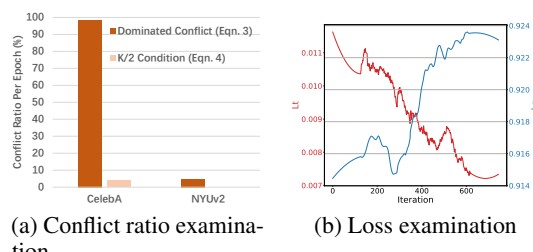

(a) Conflict ratio examination

(b) Loss examination

Figure 5: (a) Conflict ratio per epoch on CelebA (40-task) and NYUv2 (3-task) and (b) loss examinations during a single teleportation.

$$\mathcal{L}(\boldsymbol{\theta}) = \mathcal{L}(\boldsymbol{\theta} + \Delta\boldsymbol{\theta}) \tag{5}$$

$$\Delta\boldsymbol{\theta} = \underset{\Delta\boldsymbol{\theta}}{\arg\max} \|\nabla\mathcal{L}(\boldsymbol{\theta} + \Delta\boldsymbol{\theta})\|^2. \tag{6}$$

With respect to the specific symmetry teleportation taking place during the optimization process, in order to ensure the task loss remains invariant, we undertake the minimization of the loss fluctuation in the following way:

$$\mathcal{L}_t = \frac{1}{K} \sum_{i}^{K} |\mathcal{L}_i - \mathcal{L}_i^*| \tag{7}$$

where $\mathcal{L}_i$ represents the individual task loss, and $\mathcal{L}_i^*$ is its loss before starting teleportation, which is unchanged during teleportation.

---

[2] We have provided a comparison between dominated conflict and weak conflict in the **Appendix** (Sec. A.4).

To maximize the gradient of the target point, a simplistic solution would be to incorporate it into the objective of LoRA optimization. However, two challenges arise when attempting to do so: (1) It is difficult to explicitly incorporate gradient maximization into the objective design. (2) Even if it were possible, the computation of the Hessian matrix would be overly burdensome for LoRA optimization. On the other hand, upon closely examining Eqn. 6, we can observe that our intention is merely to identify the point with the maximal gradient, rather than precisely attaining the maximal gradient itself. Consequently, we choose to select another metric for measuring the gradient norm, i.e., *Sharpness* [Foret et al., 2021]. Since the negative direction of the gradient is locally the fastest direction of descent, thus we can estimate the gradient by seeking the sharpest direction at $\theta^*$ as follow:

$$\text{Sharpness} = \max_{\|\epsilon\| \leq \delta} |\mathcal{L}(\theta^* + \epsilon) - \mathcal{L}(\theta^*)| \tag{8}$$

where $\delta$ is the radius of the sphere. We further implement Eqn. 8 by randomly sampling $\epsilon$ $\tilde{n}$ times from the sphere, and estimating sharpness by selecting the maximum one:

$$\mathcal{L}_g = \max \left\{ \left| \frac{1}{K} \sum_i^K R_i \cdot [\mathcal{L}_i(\theta + \Delta\theta + \epsilon_j) - \mathcal{L}_i(\theta + \Delta\theta)] \right| \right\}_{j=1}^{\tilde{n}} \tag{9}$$

$$\mathbf{R} = K \cdot \text{softmax}\left( \left[ \frac{\sum_{j=1}^K \|g_j\|}{\|g_i\|} \right]_{i=1}^K \right) \tag{10}$$

where $\mathbf{R}$ is computed to facilitate the search for more balanced alternatives, mitigating imbalance issues. Consequently, the overall objective for LoRA optimization can be formulated as follows:

$$\mathcal{L}_{lora} = \mathcal{L}_t - \gamma \mathcal{L}_g \tag{11}$$

where $\gamma$ is the hyper-parameter (set as 0.1). As depicted in Figure 5(b), $\mathcal{L}_t$ largely decreases while $\mathcal{L}_g$ increases as expected during teleportation.

To enhance understanding of our approach, we provide a trajectory illustration on toy examples [Liu et al., 2021a] in Figure 6. As shown, LS may fail to reach the Pareto front from certain initializations due to conflict issues. However, with the augmentation of COST, it successfully explores alternative paths for continuous optimization rather than getting stuck.

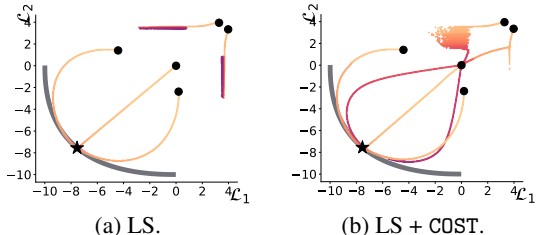

(a) LS.  (b) LS + COST.

Figure 6: Trajectory illustration on toy examples.

## 4.2 Convergence Analysis

In this section, we present a convergence analysis to further enhance the understanding of the applicability of our proposed method. By formulating a theorem, it has been proven that our method converges to the Pareto stationary point with guarantee.

**Theorem 2.** *Assume task loss functions $\mathcal{L}_1, ..., \mathcal{L}_K$ are differentiable and $\Lambda$-smooth ($\Lambda > 0$) such that $\|\nabla \mathcal{L}_i(\theta_1) - \nabla \mathcal{L}_i(\theta_2)\| \leq \Lambda \|\theta_1 - \theta_2\|$ for any two points $\theta_1$, $\theta_2$, and our symmetry teleportation property holds. Set the step size as $\eta = \frac{1}{\Lambda\sqrt{T-1}}$, $T$ is the training iteration. Then, there exists a subsequence $\{\theta^{t_j}\}$ of the output sequence $\{\theta^t\}$ that converges to a Pareto stationary point $\theta^*$.*

The proof of this theorem is provided in the **Appendix** (Sec. F).

## 4.3 Historical Trajectory Reuse Strategy

When training MTL models with advanced optimizers, a minor issue arises after reaching the loss-invariant point through our symmetry teleportation. Specifically, the teleportation process disrupts the continuous optimization flow, preventing the MTL model from leveraging its historical trajectory—one of the key advantages of advanced optimizers (e.g., Adam [Kingma, 2014]).

Taking the Adam optimizer as an example, which is commonly utilized in mainstream MTL approaches [Liu et al., 2021a, 2024a, Navon et al., 2022]. It employs an exponentially weighted moving average to estimate the momentum ($v_t$) and quadratic moments ($s_t$) of the gradient (historical trajectory). However, when the model is teleported to another point, the stored historical trajectory would supply misleading information for the current model optimization. To tackle this issue, we partially preserve the historical trajectory by computing the correlation between teleportation and previous updating (as depicted in Figure 7):

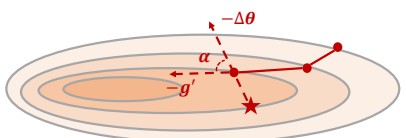

Figure 7: The illustration of HTR strategy. The red star represents the teleported point. $g'$ is the gradient at the pre-teleportation point, and $\alpha$ is the angle between $g'$ and $\Delta\theta$.

$$\sigma = \cos\_\text{sim}(\Delta\theta, g') \tag{12}$$

where $\cos\_\text{sim}$ is the function of computing cosine similarity, and $g'$ is the gradient at the pre-teleportation point. In this way, the historical trajectory can be modulated and reused according to Eqn. 13. After that, $\sigma$ is still set to 1, as is typically the case with the Adam optimizer.

$$v_t = \sigma\beta_1 v_{t-1} + (1 - \sigma\beta_1)g_t \tag{13}$$
$$s_t = \sigma\beta_2 s_{t-1} + (1 - \sigma\beta_2)g_t^2$$

The overall training algorithm is concluded in the **Appendix** (Section C).

## 5  Performance Evaluation

In this section, we initially evaluate our method using mainstream MTL benchmarks and compare it with the following baselines: Linear Scalarization (LS), Scale-Invariant (SI), Random Loss Weighting (RLW) as described in [Lin et al., 2021], Dynamic Weight Average (DWA) from [Liu et al., 2019], Uncertainty Weighting (UW) detailed in [Kendall et al., 2018], MGDA from [Sener and Koltun, 2018], GradDrop presented in [Chen et al., 2020], PC-Grad as in [Yu et al., 2020], CAGrad from [Liu et al., 2021a], IMTL detailed in [Liu et al., 2021b], Nash-MTL from [Navon et al., 2022], FAMO described in [Liu et al., 2024a], and FairGrad ($\alpha = 1$) from [Ban and Ji, 2024]. Sub-

Table 1: **Scene understanding** (*CityScapes*, 2 tasks).

| Method | Segmentation ↑ | | Depth ↓ | | MR ↓ | Δm% ↓ |
|---|---|---|---|---|---|---|
| | mIoU | Pix. Acc. | Abs. Err. | Rel. Err. | | |
| Independent | 74.01 | 93.16 | 0.0125 | 27.77 | - | - |
| LS | 75.18 | 93.49 | 0.0155 | 46.77 | 8.25 | 22.60 |
| RLW | 74.57 | 93.41 | 0.0158 | 47.79 | 11.00 | 24.37 |
| DWA | 75.24 | 93.52 | 0.0160 | 44.37 | 8.25 | 21.43 |
| Uncertainty | 72.02 | 92.85 | 0.0140 | 30.13 | 7.50 | 5.88 |
| MGDA | 68.84 | 91.54 | 0.0309 | 33.50 | 11.00 | 44.14 |
| GradDrop | 75.27 | 93.53 | 0.0157 | 47.54 | 7.75 | 23.67 |
| PCGrad | 75.13 | 93.48 | 0.0154 | 42.07 | 8.50 | 18.21 |
| CAGrad | 75.16 | 93.48 | 0.0141 | 37.60 | 7.50 | 11.58 |
| IMTL | 75.33 | 93.49 | 0.0135 | 38.41 | 5.75 | 11.04 |
| Nash-MTL | 75.41 | 93.66 | 0.0129 | 35.02 | 3.00 | 6.82 |
| FAMO | 74.54 | 93.29 | 0.0145 | 32.59 | 8.25 | 8.13 |
| FairGrad | 75.72 | 93.68 | 0.0134 | 32.25 | 2.25 | 5.18 |
| COST | 75.73 | 93.53 | 0.0133 | 31.53 | 2.00 | 4.30$_{\pm 1.3}$ |

sequently, we offer some additional analyses regarding conflict and gradient examinations, ablation studies, and plug-and-play verification, etc., to further enhance understanding. We also provide additional analysis on conflict and gradient examination (Sec. A.1), plug-and-play verification (Sec. A.2), alternatives to PEFT (Sec. A.5), and time cost (Sec. A.6) in the **Appendix**. All experiments are carried out on a single Tesla V100 GPU. For more experimental details, please refer to the **Appendix** (Sec. B). Code will be released once this paper is accepted.

**Evaluation Metric**. In addition to reporting individual performance, we also incorporate a widely used metric, **Δm%** [Maninis et al., 2019], which evaluates the overall degradation compared to independently trained models that are considered as the reference oracles. The formal definition of Δm% is given as:

$$\Delta\mathbf{m\%} = \frac{1}{K}\sum_{k=1}^{K}(-1)^{\delta_k}\frac{M_{m,k} - M_{b,k}}{M_{b,k}} \times 100 \tag{14}$$

where $M_{m,k}$ and $M_{b,k}$ represent the metric $M_k$ for the compared method and the independent model, respectively. The value of $\delta_k$ is assigned as 1 if a higher value is better for $M_k$, and 0 otherwise. Besides, we also report another popular metric named **Mean Rank** (**MR**), which computes the average ranks of each methods across all tasks.

## 5.1 Overall Evaluation

**Dense Prediction.** CityScapes [Cordts et al., 2016] and NYUv2 [Silberman et al., 2012] are two widely-used scene understanding datasets, which are employed for the evaluation of MTL. NYUv2 comprises 1449 annotated images and is utilized for three fine-grained tasks, i.e., semantic segmentation, depth estimation, and surface normal prediction. CityScapes consists of 5000 annotated scene images, which are readied for two tasks: semantic segmentation and depth estimation.

Table 2: **Scene understanding** (*NYUv2*, 3 tasks). We report MTAN model performance averaged over 3 random seeds. The best scores are provided in  gray , and the second scores are underlined.

| Method | Segmentation ↑ | | Depth ↓ | | Surface Normal | | | | | MR ↓ | Δm% ↓ |
|---|---|---|---|---|---|---|---|---|---|---|---|
| | | | | | Angle Distance ↓ | | Within $t°$ ↑ | | | | |
| | mIoU | Pix. Acc. | Abs. Err. | Rel. Err. | Mean | Median | 11.25 | 22.5 | 30 | | |
| Independent | 38.30 | 63.76 | 0.68 | 0.28 | 25.01 | 19.21 | 30.14 | 57.20 | 69.15 | - | - |
| LS | 39.29 | 65.33 | 0.55 | 0.23 | 28.15 | 23.96 | 22.09 | 47.50 | 61.08 | 9.44 | 5.46 |
| RLW | 37.17 | 63.77 | 0.58 | 0.24 | 28.27 | 24.18 | 22.26 | 47.05 | 60.62 | 12.22 | 7.67 |
| DWA | 39.11 | 65.31 | 0.55 | 0.23 | 27.61 | 23.18 | 24.17 | 50.18 | 62.39 | 8.56 | 3.49 |
| Uncertainty | 36.87 | 63.17 | 0.54 | 0.23 | 27.04 | 22.61 | 23.54 | 49.05 | 63.65 | 8.78 | 4.01 |
| MGDA | 30.47 | 59.90 | 0.61 | 0.26 | 24.88 | 19.45 | 29.18 | 56.88 | 69.36 | 7.11 | 1.47 |
| GradDrop | 39.39 | 65.12 | 0.55 | 0.23 | 27.48 | 22.96 | 23.38 | 49.44 | 62.87 | 8.89 | 3.61 |
| PCGrad | 38.06 | 64.64 | 0.56 | 0.23 | 27.41 | 22.80 | 23.86 | 49.83 | 63.14 | 9.33 | 3.83 |
| CAGrad | 39.79 | 65.49 | 0.55 | 0.23 | 26.31 | 21.58 | 25.61 | 52.36 | 65.58 | 6.33 | 0.29 |
| IMTL | 39.35 | 65.60 | 0.54 | 0.23 | 26.02 | 21.19 | 26.20 | 53.13 | 66.24 | 5.56 | -0.59 |
| Nash-MTL | 40.13 | 65.93 | 0.53 | 0.22 | 25.26 | 20.08 | 28.40 | 55.47 | 68.15 | 3.11 | -4.04 |
| FAMO | 40.30 | 66.07 | 0.56 | 0.21 | 26.67 | 21.83 | 25.61 | 51.78 | 64.85 | 5.44 | 0.16 |
| FairGrad | 39.74 | 66.01 | 0.54 | 0.22 | 24.84 | 19.60 | 29.26 | 56.58 | 69.16 | 3.00 | -4.66 |
| COST | 38.06 | 64.71 | 0.54 | 0.23 | 24.47 | 18.80 | 30.84 | 58.25 | 70.30 | 3.22 | $-5.39_{\pm 0.5}$ |

In line with the implementation and training strategy of FairGrad [Ban and Ji, 2024], we construct our model using SegNet [Badrinarayanan et al., 2017] and employ MTAN [Liu et al., 2019] as the backbone within it. We train our model with the Adam optimizer for a total of 200 epochs, setting the initial learning rate to 1.0e-4 and reducing it to half after 100 epochs. The batch size is set to 2 for NYUv2 and 8 for CityScapes, respectively.

The results obtained on these two datasets are presented in Table 1 and Table 2, respectively. With FairGrad serving as the baseline, our method not only successfully surpasses it but also attains the SOTA performance in terms of **MR** and Δm%. Specifically, upon closely examining the performance of each individual task, we can note that COST significantly enhances FairGrad on the CityScapes dataset and considerably improves the surface normal prediction task, while also showing some promise on the other tasks on the NYUv2 dataset. These observations clearly demonstrate the effectiveness of our design, which aids in alleviating conflict and facilitating convergence.

**Image Classification.** CelebA [Liu et al., 2015] is a commonly utilized face attributes dataset [Wang et al., 2024] that contains over 200,000 images and is annotated with 40 attributes. Recently, it has been adopted as a 40-task MTL benchmark to assess the model's capacity to handle a large number of tasks. In accordance with the setup of FairGrad, we utilize a 9-layer convolutional neural network (CNN) as the backbone and linear layers as the task-specific heads on

Table 3: Results on *CelebA* and *QM9* datasets with MR and Δm%. The results of FairGrad-R are reported according to the official implementation.

| Method | CelebA | | QM9 | |
|---|---|---|---|---|
| | MR ↓ | Δm% ↓ | MR ↓ | Δm% ↓ |
| LS | 7.08 | 4.15 | 9.09 | 177.6 |
| SI | 8.80 | 7.20 | 5.64 | 77.8 |
| RLW | 5.98 | 1.46 | 10.64 | 203.8 |
| DWA | 7.78 | 2.40 | 8.91 | 175.3 |
| UW | 6.65 | 3.23 | 7.00 | 108.0 |
| MGDA | 11.98 | 14.85 | 8.91 | 120.5 |
| PCGrad | 7.58 | 3.17 | 7.36 | 125.7 |
| CAGrad | 7.13 | 2.48 | 8.09 | 112.8 |
| IMTL-G | 5.53 | 0.84 | 6.91 | 77.2 |
| Nash-MTL | 5.73 | 2.84 | 4.27 | 62.0 |
| FAMO | 5.65 | 1.21 | 5.18 | 58.5 |
| FairGrad-R | 6.35 | 1.15 | 4.82 | 59.9 |
| COST | 4.80 | $0.93_{\pm 0.2}$ | 4.18 | $58.3_{\pm 1.2}$ |

top of it. We train our model with the Adam optimizer for a total of 15 epochs, setting the initial learning rate to 3.0e-4. Moreover, the batch size is set to 256.

The evaluation results are shown in Table 3. Given that our method is mainly developed based on FairGrad, our performance is thus highly associated with it. We conscientiously re-implemented FairGrad using the official code they provided and were able to achieve the reported performance on CityScapes, NYUv2. However, we were unable to do so on CelebA and QM9. Consequently, we only report our re-implemented performance of FairGrad here (referred to as FairGrad-R). As can be observed, COST still significantly enhances its baseline and attains the SOTA performance according to MR, ranking second according to $\Delta$m%. These results demonstrate COST's remarkable ability to handle numerous tasks simultaneously.

**Regression.** QM9 [Ramakrishnan et al., 2014] is another widely used drug discovery MTL dataset specifically for regression tasks. It contains 130,000 organic molecules that are organized as graphs with node and edge features. This task is designed to predict 11 properties having different measurement scales and can also be considered as an evaluation scenario for MTL involving a large number of tasks. Our approach is trained for 300 epochs with a batch size of 120. The initial learning rate is set to 1.0e-3, and a learning rate scheduler is applied to reduce the rate when the validation performance shows no further improvement.

According to Table 3, our method still achieves competitive performance on this specific dataset. However, in comparison to other datasets, it exhibits fewer enhancements over its baseline. One crucial reason for these relatively less satisfactory results is that this task adopts a graph model with only two layers supporting LoRA in current PEFT package, which reduces its effectiveness.

## 5.2 Ablation Study

We consider COST as an integrated system, and thus each component ought to be evaluated to showcase its effectiveness. In our design, there are primarily three key components: the loss invariance objective ($\mathcal{L}_t$), the gradient maximization objective ($\mathcal{L}_g$), and the HTR strategy. Consequently, we carry out ablation studies for verification purposes and present the results in Table 4. In the context of symmetry teleportation, $\mathcal{L}_t$ and $\mathcal{L}_g$ serve as the foundation for seeking alternatives within the orbit. Hence, we exclude $\mathcal{L}_t$ and $\mathcal{L}_g$ during the LoRA optimization process, respectively. The results indicate that without $\mathcal{L}_t$ or $\mathcal{L}_g$,

Table 4: Ablation study of COST on *CityScapes* (2 tasks).

| $\mathcal{L}_t$ | $\mathcal{L}_g$ | HTR | $\Delta$m% $\downarrow$ |
|---|---|---|---|
|  |  |  | 5.18 |
| ✓ |  |  | 7.90 |
| - | ✓ |  | **381.86** |
| ✓ | ✓ |  | 4.65 |
| ✓ | ✓ | ✓ | 4.30 |

COST performs worse than its baseline (which is FairGrad in this case). More specifically, COST would experience a severe deterioration without $\mathcal{L}_t$, thereby underlining the crucial importance of loss invariance. These results might address another concern regarding COST, namely: Are the improvements brought about by COST rooted in the capability expansion facilitated by LoRA? Without an appropriate objective design, LoRA is unable to effectively augment the base models.

On the other hand, it should be noted that LoRA is incorporated into the base model after each teleportation. Thus, the model's capability remains unchanged during the inference time. All that we are doing is assisting in finding a better convergence point. Furthermore, when the HTR strategy is excluded, $\Delta$m% decreases from 4.30 to 4.65, which demonstrates the significance of benefiting from advanced optimizers.

# 6 Conclusion

This paper explores the MTL problem from a brand new perspective, i.e., alleviating the conflict issue through symmetry teleportation. Specifically, we utilize LoRA to achieve practical symmetry teleportation for contemporary deep models. Additionally, we design loss-invariant and gradient maximization objectives to assist in identifying non-conflict and more convergent points. We also devise a historical trajectory reuse strategy to continuously benefit from advanced optimizers. Extensive experiments have demonstrated the effectiveness of our proposed method as well as its plug-and-play characteristic. As a scalable framework, we anticipate that our method can offer some

valuable insights to researchers engaged in optimization-based MTL. Currently, there are still rooms for improvement within this system, and our future work will focus on these aspects.

## Acknowledgement

This research is supported, in part, by the WeBank-NTU Joint Research Institute on Fintech, Jinan-NTU Green Technology Research Institute (GreenTRI), and Joint NTU-UBC Research Centre of Excellence in Active Living for the Elderly (LILY), Nanyang Technological University, Singapore. It's also supported, in part, by the NTU-PKU Joint Research Institute, a collaboration between Nanyang Technological University and Peking University that is sponsored by a donation from the Ng Teng Fong Charitable Foundation. The first author also would like to thank the insightful discussion with Dr. Shuaicheng Niu, Shibo Feng, and Haochen Li.

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

# A   Additional Results

## A.1   Conflict and Gradient Examinations

Although our method achieves competitive performance, it remains unclear whether it effectively resolves the targeted issues, i.e., conflict mitigation and greater gradient norm discovery. To investigate this, we analyze the training process by recording the results before and after teleportation, as shown in Figure 8. The findings indicate that conflict is significantly alleviated, with task gradients becoming positively correlated in most cases after teleportation. Besides, teleportation consistently yields greater gradient norms, confirming the effectiveness of COST's design.

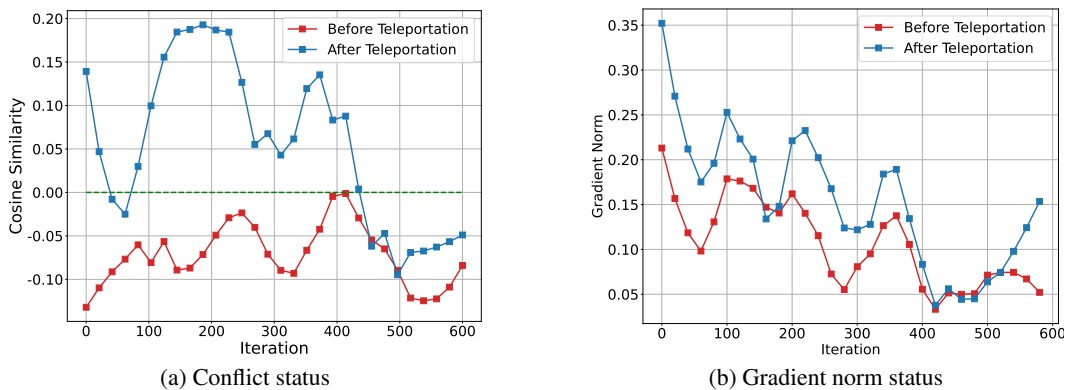

(a) Conflict status             (b) Gradient norm status

Figure 8: Examinations before/after teleportation.

## A.2   Plug-and-Play Verification

Intuitively, our method is orthogonal to existing MTL approaches and is therefore plug-and-play, enabling augmentation when integrated. Here, we take three baselines (i.e., CAGrad, Nash-MTL, and FairGrad) to demonstrate the effectiveness of COST, and present the results in Table 5. As anticipated, our method successfully brings considerable augmentation to its baselines, with improvements ranging from 0.88 to 3.21 according to $\Delta$m%. Specifically, CAGrad and FairGrad receive improvements on almost each individual metric.

We also conduct an additional verification experiment using CAGrad on the NYUv2 dataset. The results, presented in Table 6, show that COST consistently enhances CAGrad's performance across all metrics.

Table 5: Plug-and-play verification on *CityScapes* (2 tasks) dataset. We adopt FAMO's implementation for Nash-MTL (denoted as Nash-R) and augment it with COST, since Nash-MTL does not provide the official implementation on CityScapes.

| Method | Segmentation ↑ | | Depth ↓ | | $\Delta$m% ↓ |
|---|---|---|---|---|---|
| | mIoU | Pix. Acc. | Abs. Err. | Rel. Err. | |
| CAGrad | 75.16 | 93.48 | 0.0141 | 37.60 | 11.58 |
| CAGrad + COST | 75.46 | 93.57 | 0.0134 | 35.68 | 8.37 |
| Nash-R | 75.87 | 93.57 | 0.0135 | 37.29 | 9.89 |
| Nash-R + COST | 75.70 | 93.56 | 0.0134 | 34.34 | 7.15 |
| FairGrad | 75.72 | 93.68 | 0.0134 | 32.25 | 5.18 |
| FairGrad + COST | 75.73 | 93.53 | 0.0133 | 31.53 | 4.30 |

Table 6: Plug-and-play verification on *NYUv2* (3 tasks).

| Method | Segmentation | | Depth | | Surface Normal | | | | | $\Delta$m% $\downarrow$ |
|---|---|---|---|---|---|---|---|---|---|---|
| | (Higher Better) | | (Lower Better) | | Angle Distance | | Within $t°$ | | | |
| | | | | | (Lower Better) | | (Higher Better) | | | |
| | mIoU | Pix. Acc. | Abs Err | Rel Err | Mean | Median | 11.25 | 22.5 | 30 | |
| CAGrad | 39.79 | 65.49 | 0.55 | 0.23 | 26.31 | 21.58 | 25.61 | 52.36 | 65.58 | 0.29 |
| CAGrad + COST | 40.76 | 66.42 | 0.53 | 0.22 | 26.00 | 21.13 | 26.44 | 53.25 | 66.30 | -1.61 |
| FairGrad | 39.74 | 66.01 | 0.54 | 0.22 | 24.84 | 19.60 | 29.26 | 56.58 | 69.16 | -4.66 |
| FairGrad + COST | 38.06 | 64.71 | 0.54 | 0.23 | 24.47 | 18.80 | 30.84 | 58.25 | 0.30 | -5.39 |

## A.3 Motivation

To further elucidate our motivation, we additionally carry out a verification experiment to observe the dominant conflict status within Nash-MTL. Based on the results shown in Figure 9, Nash-MTL exhibits fewer dominated conflicts in comparison to CAGrad and FairGrad, yet still encounters a significant number. Moreover, it also possesses symmetry points that have the same loss level but with different conflict status.

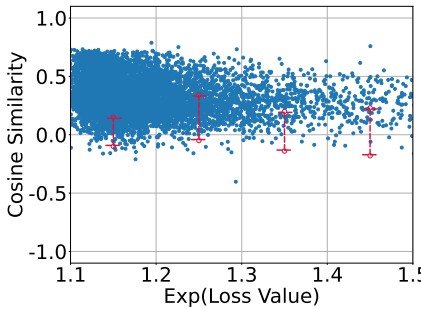

Figure 9: Dominated conflict vs. loss examination of Nash-MTL.

## A.4 Analysis on Trigger Condition

As stated in the main text, the trigger condition is of importance for our symmetry teleportation process. To illustrate this, we carry out a comparison between our system when triggered by dominated conflict and when triggered by weak conflict, and the results are presented in Table 7. It can be observed that although COST-weak surpasses COST-dominated in terms of most individual metrics, it accomplishes this by sacrificing a significant portion of the performance on the Rel. Err. metric. This is an unexpected outcome in the context of MTL. Consequently, it does not perform satisfactorily on the overall metric ($\Delta$m%). These results further emphasize that addressing both imbalance and conflict issues is crucial for MTL. This is because the scenario of weak conflict only focuses on the conflict issue, overlooking the importance of handling imbalance as well.

Table 7: Trigger condition comparison on *CityScapes* (2 tasks) dataset.

| Method | Segmentation | | Depth | | $\Delta$m% $\downarrow$ |
|---|---|---|---|---|---|
| | (Higher Better) | | (Lower Better) | | |
| | mIoU | Pix. Acc. | Abs. Err. | Rel. Err. | |
| COST -dominated | 75.73 | 93.53 | 0.0133 | 31.53 | 4.30 |
| COST -weak | 75.92 | 93.64 | 0.0127 | 35.94 | 6.94 |

We also conduct an additional experiment on the trigger condition in the presence of numerous tasks, following Eqn. 4 in the main text. As previously stated, this condition is designed to balance

effectiveness and efficiency—relaxing it would improve performance but at the cost of increased inefficiency. The results, presented in Figure 10, confirm this trade-off. As shown, $\Delta$m% gradually decreases as the trigger condition is relaxed, with improvements driven by more frequent teleportation. However, this comes at the cost of an almost linear increase in time complexity.

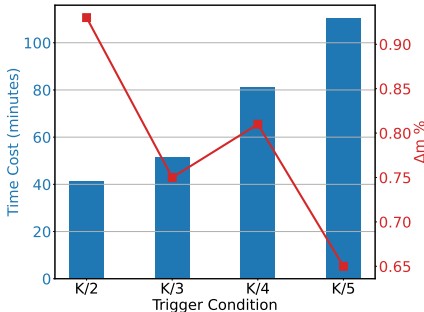

Figure 10: Trigger condition under the massive tasks scenario. Time cost is measured for one epoch on CelebA.

Table 8: PEFT alternatives comparison on *CityScapes* (2 tasks) dataset.

| Method | Segmentation | | Depth | | $\Delta$m% ↓ |
|---|---|---|---|---|---|
| | (Higher Better) | | (Lower Better) | | |
| | mIoU | Pix. Acc. | Abs. Err. | Rel. Err. | |
| FairGrad | 75.72 | 93.68 | 0.0134 | 32.25 | 5.18 |
| COST-LoRA | 75.73 | 93.53 | 0.0133 | 31.53 | 4.30 |
| COST-LoHa Hyeon-Woo et al. [2021] | 75.52 | 93.43 | 0.0130 | 35.07 | 7.10 |
| COST-OFT Qiu et al. [2023] | 68.17 | 91.40 | 0.0151 | 45.25 | 23.39 |

## A.5 Analysis on Alternative of PEFT

Currently, numerous PEFT alternatives are available for teleportation purposes. To further examine the effect of PEFT on our method, we employ additional PEFT options to assess their impact on MTL performance. Specifically, we evaluate a LoRA variant (LoHa) and another PEFT alternative, OFT. The results, presented in Table 8, show that neither PEFT option improves upon their baselines. This suggests that while advanced PEFT methods may enable more efficient tuning, their complex designs can limit generalizability across various scenarios, aligning with some recent observations [Pu et al., 2023]. Identifying a suitable PEFT approach remains a future direction for our framework.

The obtained results demonstrate that our framework can indeed benefits from certain other PEFT alternatives, e.g., LoHa. However, it also encounters setbacks when using alternatives like OFT. This implies that the selection of the PEFT method warrants further investigation.

## A.6 Time Cost

Applying teleportation at every instance of a conflict would significantly increase the computational burden and training time. To mitigate this, we introduce two strategies: delayed start and frequency control. The delayed start strategy postpones the application of teleportation until after $E$ epochs. Meanwhile, frequency control limits the number of teleportation operations within each epoch, reducing overhead without much compromising the optimization process.

Here, we measure the running time of a single epoch on CelebA, comparing the scenarios with and without the COST augmentation, and present the results in Figure 11. As can be observed, our applied strategies introduce only an additional 30% of the training time compared to its baselines. Nonetheless, we acknowledge this still constitutes one of our limitations.

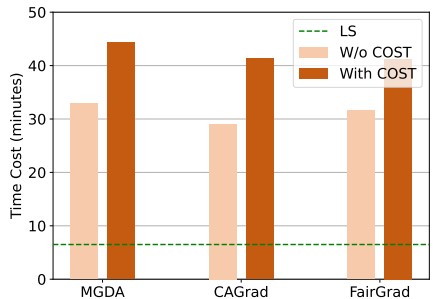

Figure 11: Time cost comparison.

# B   Implementation Details

## B.1   Baseline Implementation

**CAGrad**: CAGrad strikes a balance between Pareto optimality and globe convergence by regulating the combined gradients in proximity to the average gradient:

$$\max_{\boldsymbol{d}\in\mathbb{R}^m}\min_{\boldsymbol{\omega}\in\mathcal{W}}\boldsymbol{g}_{\boldsymbol{\omega}}^\top\boldsymbol{d} \quad \text{s.t.}\,\|\boldsymbol{d}-\boldsymbol{g_0}\| \leq c\,\|\boldsymbol{g_0}\| \tag{15}$$

In our experiments, we extend the official implementation to include `COST`. And all our training settings follow the same manner with its original one. For more comprehensive information, please consult the official implementation.

**Nash-MTL**: Nash-MTL provides the individual progress guarantee via the following objective:

$$\min_{\boldsymbol{\omega}}\sum_i \beta_i(\boldsymbol{\omega}) + \varphi_i(\boldsymbol{\omega}) \tag{16}$$

$$s.t.\forall i, -\varphi_i(\boldsymbol{\omega}) \leq 0, \quad \omega_i > 0. \tag{17}$$

where $\varphi_i(\boldsymbol{\omega}) = \log(\omega_i) + \log(\boldsymbol{g_i}^\top\boldsymbol{G\omega})$, $\boldsymbol{G} = [\boldsymbol{g_1}, \boldsymbol{g_2}, ..., \boldsymbol{g_K}]$. As demonstrated, the individual progress is ensured through the projection, subject to the constraint $\beta_i = \boldsymbol{g_i}^\top\boldsymbol{G\omega} \geq \frac{1}{\omega_i}$.

We utilize the FAMO's implementation of Nash-MTL, since Nash-MTL's official code does not provide the implementation on CityScapes (2 tasks). In the Table 4 of the main text, we honestly report the results of this implementation, and show improvements brought by `COST`.

**FairGrad**: FairGrad is a pioneer MTL algorithm that proposes fairness measurements to promote maximal loss decrease, and formulate the following objective to derive the combination of individual gradients:

$$\boldsymbol{G}^\top\boldsymbol{G\omega} = \boldsymbol{\omega}^{-1/\alpha} \tag{18}$$

where $\alpha$ is the hyper-parameter, which is set to 1. We regard FairGrad as the advanced version of Nash-MTL that is able to balance task progresses in a finer grained. We adopt its official implementation for all our implementation through the paper.

## B.2   PEFT Implementation

**LoHa**: As a variant of LoRA, it approximates large weight matrices with low-rank ones through the Hadamard product. This approach has the potential to be more parameter-efficient than LoRA itself.

**OFT**: OFT draws inspiration from continual learning. It operates by re-parameterizing the pre-trained weight matrices using its orthogonal matrix, thereby preserving the information within the pre-trained model. To decrease the number of parameters, OFT incorporates a block-diagonal structure into the orthogonal matrix.

We utilize the implementations of LoRA, LoHa, and OFT provided by Hugging Face's PEFT. **The rank for each of these is set to 5, while other configurations are left at their default values**. We apply LoRA to the backbone network across three datasets (CityScapes, NYUv2, and CelebA). For the QM9 benchmark, since it employs a graph network which is currently not supported by PEFT, we only apply LoRA to input and output linear layers instead.

---

**Algorithm 1** `COST` for MTL

---

Model parameters $\boldsymbol{\theta^0}$
Initialize Initialize $\boldsymbol{\theta^0}$ randomly
$t = 1$ to $T$ Compute task gradients $G = [\boldsymbol{g_i}]_{i=1}^K$
Dominated conflict detected Freeze $\boldsymbol{\theta^t}$ and train LoRA ($\boldsymbol{\Delta\theta^t}$) according Eqn.7, 9, 10, and 11 in the main text;
Merge $\boldsymbol{\theta^t}$ and $\boldsymbol{\Delta\theta^t}$: $\boldsymbol{\theta^t} = \boldsymbol{\theta^t} + \boldsymbol{\Delta\theta^t}$, and unfreeze $\boldsymbol{\theta^t}$
Apply HTR on the optimizer according to Eqn.12, and 13 in the main text;  Other MTL optimization
Have applied HTR Reset $\sigma$ to 1;

---

## C  Algorithm

We conclude the learning paradigm of `COST` in Algorithm 1. It should be noted that since `COST` is a scalable framework, thus the other MTL optimization in Algorithm 1 could be mainstream MTL approaches (e.g., CAGrad, Nash-MTL, and FairGrad, etc).

## D  Discussion & Limitation

We offer a new framework to address the challenges of MTL, which is highly scalable and can be further improved by integrating more advanced components. For instance, LoRA could be substituted with an alternative PEFT method, and sharpness estimation can be substituted with some efficient gradient estimation methods [Liu et al., 2024b]. However, there are still some limitations. The current training paradigm requires additional training costs, and its performance on regression tasks is less competitive compared to the others. We have discussed some of these limitations in the Appendix, while others are left for future work.

## E  Pareto Concept

Formally, let us assume the weighted loss as $\mathcal{L}_{\boldsymbol{\omega}} = \sum_{i=1}^K \omega_i \mathcal{L}_i(\boldsymbol{\theta})$, where $\boldsymbol{\omega} \in \mathcal{W}$ and $\mathcal{W}$ represents the probability simplex on $[K]$. A point $\boldsymbol{\theta'}$ is said to Pareto dominate $\boldsymbol{\theta}$ if and only if $\forall i, \mathcal{L}_i(\boldsymbol{\theta'}) \leq \mathcal{L}_i(\boldsymbol{\theta})$. Consequently, the Pareto optimal situation arises when no $\boldsymbol{\theta'}$ can be found that satisfies $\forall i, \mathcal{L}_i(\boldsymbol{\theta'}) \leq \mathcal{L}_i(\boldsymbol{\theta})$ for the given point $\boldsymbol{\theta}$. All points that meet these conditions are referred to as Pareto sets, and their solutions are known as Pareto fronts. Another concept, known as Pareto stationary, requires $\min_{\boldsymbol{\omega} \in \mathcal{W}} \|\boldsymbol{g_{\omega}}\| = 0$, where $\boldsymbol{g_{\omega}}$ represents the weighted gradient $\boldsymbol{\omega}^{\top} G$, and $G$ is the gradients matrix whose each row is an individual gradient. We also provide the definition of gradient similarity for ease of description.

## F  Convergence Analysis

**Lemma 3.** *Let $\mathcal{L}(\boldsymbol{\theta}, \xi)$ be a $\Lambda$-smooth function, where $\xi$ is the i.i.d sampled mini-batch data. It follows that:*

$$\mathbb{E}\left[\|\nabla \mathcal{L}(\boldsymbol{\theta}, \xi)\|^2\right] \leq$$
$$2\Lambda\left(\mathcal{L}(\boldsymbol{\theta}) - \mathcal{L}(\boldsymbol{\theta^*})\right) + 2\Lambda\left(\mathcal{L}(\boldsymbol{\theta^*}) - \mathbb{E}\left[\inf_{\boldsymbol{\theta}} \mathcal{L}(\boldsymbol{\theta}, \xi)\right]\right)$$

*Proof.* We have the following inequality according to the $\Lambda$-smooth property of $\mathcal{L}(\boldsymbol{\theta}, \xi)$:

$$\mathcal{L}(\boldsymbol{\theta'}, \xi) - \mathcal{L}(\boldsymbol{\theta}, \xi) \leq \tag{19}$$
$$\langle \nabla \mathcal{L}(\boldsymbol{\theta}, \xi), \boldsymbol{\theta'} - \boldsymbol{\theta} \rangle + \frac{\Lambda}{2} \|\boldsymbol{\theta'} - \boldsymbol{\theta}\|^2, \forall \boldsymbol{\theta'}, \boldsymbol{\theta} \in \mathbb{R}^d$$

And $\boldsymbol{\theta'} = \boldsymbol{\theta} - \frac{1}{\Lambda} \nabla \mathcal{L}(\boldsymbol{\theta}, \xi)$, thus we have:

$$\mathcal{L}(\boldsymbol{\theta} - (1/\Lambda)\nabla \mathcal{L}(\boldsymbol{\theta}, \xi), \xi) \leq \mathcal{L}(\boldsymbol{\theta}, \xi) - \frac{1}{2\Lambda} \|\nabla \mathcal{L}(\boldsymbol{\theta}, \xi)\|^2 \tag{20}$$

Assume $\boldsymbol{\theta}^* = \arg\min_{\boldsymbol{\theta}\in\mathbb{R}^d} \mathcal{L}(\boldsymbol{\theta})$, then we can re-arranging the above inequality to have:

$$\mathcal{L}(\boldsymbol{\theta}^*,\xi) - \mathcal{L}(\boldsymbol{\theta},\xi) = \tag{21}$$
$$\mathcal{L}(\boldsymbol{\theta}^*,\xi) - \inf_{\boldsymbol{\theta}} \mathcal{L}(\boldsymbol{\theta},\xi) + \inf_{\boldsymbol{\theta}} \mathcal{L}(\boldsymbol{\theta},\xi) - \mathcal{L}(\boldsymbol{\theta},\xi)$$

$$\leq \mathcal{L}(\boldsymbol{\theta}^*,\xi) - \inf_{\boldsymbol{\theta}} \mathcal{L}(\boldsymbol{\theta},\xi) + \mathcal{L}(\boldsymbol{\theta} - \frac{1}{\Lambda}\nabla\mathcal{L}(\boldsymbol{\theta},\xi),\xi) - \mathcal{L}(\boldsymbol{\theta},\xi)$$

$$\leq \mathcal{L}(\boldsymbol{\theta}^*,\xi) - \inf_{\boldsymbol{\theta}} \mathcal{L}(\boldsymbol{\theta},\xi) - \frac{1}{2\Lambda} \|\nabla\mathcal{L}(\boldsymbol{\theta},\xi)\|^2$$

where the first inequality holds because $\inf_{\boldsymbol{\theta}} \mathcal{L}(\boldsymbol{\theta},\xi) \leq \mathcal{L}(\boldsymbol{\theta},\xi), \forall\boldsymbol{\theta}$. Taking expectation on above gives:

$$\mathbb{E}\left[\|\nabla\mathcal{L}(\boldsymbol{\theta},\xi)\|^2\right]$$

$$\leq 2\mathbb{E}\left[\Lambda\left(\mathcal{L}(\boldsymbol{\theta}^*,\xi) - \inf_{\boldsymbol{\theta}} \mathcal{L}(\boldsymbol{\theta},\xi) + \mathcal{L}(\boldsymbol{\theta},\xi) - \mathcal{L}(\boldsymbol{\theta}^*,\xi)\right)\right]$$

$$\leq 2\Lambda\mathbb{E}\left[\mathcal{L}(\boldsymbol{\theta}^*,\xi) - \inf_{\boldsymbol{\theta}} \mathcal{L}(\boldsymbol{\theta},\xi) + \mathcal{L}(\boldsymbol{\theta},\xi) - \mathcal{L}(\boldsymbol{\theta}^*,\xi)\right]$$

$$\leq 2\Lambda\left(\mathcal{L}(\boldsymbol{\theta}) - \mathcal{L}(\boldsymbol{\theta}^*)\right) + 2\Lambda\left(\mathcal{L}(\boldsymbol{\theta}^*) - \mathbb{E}\left[\inf_{\boldsymbol{\theta}} \mathcal{L}(\boldsymbol{\theta},\xi)\right]\right)$$

$$\square$$

**Theorem 4.** *Assume task loss functions $\mathcal{L}_1, ..., \mathcal{L}_K$ are differentiable and $\Lambda$-smooth ($\Lambda > 0$) such that $\|\nabla\mathcal{L}_i(\boldsymbol{\theta_1}) - \nabla\mathcal{L}_i(\boldsymbol{\theta_2})\| \leq \Lambda\|\boldsymbol{\theta_1} - \boldsymbol{\theta_2}\|$ for any two points $\boldsymbol{\theta_1}$, $\boldsymbol{\theta_2}$, and our symmetry teleportation property holds. Set the step size as $\eta = \frac{1}{\Lambda\sqrt{T-1}}$, $T$ is the training iteration. Then, there exists a subsequence $\{\boldsymbol{\theta^{t_j}}\}$ of the output sequence $\{\boldsymbol{\theta^t}\}$ that converges to a Pareto stationary point $\boldsymbol{\theta^*}$.*

*Proof.* We have the following inequality according to the $\Lambda$-smooth property of $\mathcal{L}(\boldsymbol{\theta})$:

$$\mathcal{L}(\boldsymbol{\theta}') - \mathcal{L}(\boldsymbol{\theta}) \leq \langle\nabla\mathcal{L}(\boldsymbol{\theta}), \boldsymbol{\theta}' - \boldsymbol{\theta}\rangle + \frac{\Lambda}{2}\|\boldsymbol{\theta}' - \boldsymbol{\theta}\|^2 \tag{22}$$

Let $\boldsymbol{\theta}' = \boldsymbol{\theta^{t+1}}$, $\boldsymbol{\theta^{t'}} = \boldsymbol{\theta^t} + \boldsymbol{\Delta\theta^t}$ (Gradient maximization: $\boldsymbol{\Delta\theta^t} = \arg\max \nabla\mathcal{L}(\boldsymbol{\theta^t} + \boldsymbol{\Delta\theta^t})$), and $\mathcal{L}(\boldsymbol{\theta^t}) = \mathcal{L}(\boldsymbol{\theta^{t'}})$ (Loss invariance), we have:

$$\mathcal{L}(\boldsymbol{\theta^{t+1}}) \leq \mathcal{L}(\boldsymbol{\theta^{t'}}) + \langle\nabla\mathcal{L}(\boldsymbol{\theta^{t'}}), \boldsymbol{\theta^{t+1}} - \boldsymbol{\theta^{t'}}\rangle \tag{23}$$

$$+ \frac{\Lambda}{2}\left\|\boldsymbol{\theta^{t+1}} - \boldsymbol{\theta^{t'}}\right\|^2 \tag{24}$$

$$= \mathcal{L}(\boldsymbol{\theta^t}) - \eta_t\langle\nabla\mathcal{L}(\boldsymbol{\theta^{t'}}), \nabla\mathcal{L}(\boldsymbol{\theta^{t'}}, \xi^t)\rangle + \frac{\Lambda\eta_t^2}{2}\left\|\nabla\mathcal{L}(\boldsymbol{\theta^{t'}}, \xi^t)\right\|^2 \tag{25}$$

Taking expectation conditioned on $\boldsymbol{\theta^t}$, we have:

$$\mathbb{E}_t\left[\mathcal{L}(\boldsymbol{\theta^{t+1}})\right] \leq \mathcal{L}(\boldsymbol{\theta^t}) - \eta_t\left\|\nabla\mathcal{L}(\boldsymbol{\theta^{t'}})\right\|^2 \tag{26}$$

$$+ \frac{\Lambda\eta_t^2}{2}\mathbb{E}_t\left[\left\|\nabla\mathcal{L}(\boldsymbol{\theta^{t'}}, \xi^t)\right\|^2\right]$$

According to Lemma 3, we have:

$$\mathbb{E}\left[\|\nabla\mathcal{L}(\boldsymbol{\theta},\xi)\|^2\right] \leq$$

$$2\Lambda\left(\mathcal{L}(\boldsymbol{\theta}) - \mathcal{L}(\boldsymbol{\theta}^*)\right) + 2\Lambda\left(\mathcal{L}(\boldsymbol{\theta}^*) - \mathbb{E}\left[\inf_{\boldsymbol{\theta}} \mathcal{L}(\boldsymbol{\theta},\xi)\right]\right) \tag{27}$$

Inserting Eqn. 27 into Eqn. 26, we have:

$$\mathbb{E}_t\left[\mathcal{L}(\boldsymbol{\theta^{t+1}})\right] \leq \mathcal{L}(\boldsymbol{\theta^t}) - \eta_t\left\|\nabla\mathcal{L}(\boldsymbol{\theta^{t'}})\right\|^2 \tag{28}$$

$$+ \Lambda^2\eta_t^2\left(\mathcal{L}(\boldsymbol{\theta^{t'}}) - \mathcal{L}(\boldsymbol{\theta}^*) + \mathcal{L}(\boldsymbol{\theta}^*) - \mathbb{E}\left[\inf_{\boldsymbol{\theta}} \mathcal{L}(\boldsymbol{\theta},\xi)\right]\right)$$

By taking full expectation and re-arranging terms, we have:

$$\eta_t \mathbb{E}\left[\left\|\nabla\mathcal{L}(\boldsymbol{\theta}^{t'})\right\|^2\right] \leq (1+\Lambda^2\eta_t^2)\mathbb{E}\left[\mathcal{L}(\boldsymbol{\theta}^t)-\mathcal{L}^*\right] \tag{29}$$
$$- \mathbb{E}\left[\mathcal{L}(\boldsymbol{\theta}^{t+1})-\mathcal{L}^*\right] + \Lambda^2\eta_t^2\sigma^2$$

where $\sigma^2 = \mathcal{L}(\boldsymbol{\theta}^*) - \mathbb{E}\left[\inf_{\boldsymbol{\theta}}\mathcal{L}(\boldsymbol{\theta},\xi)\right]$. Then we consider to introduce the re-weighting trick in [Stich, 2019]. Let $\gamma_t$ ($\gamma_t > 0$) be a sequence such that $\gamma_t(1+\Lambda^2\eta_t^2) = \gamma_{t-1}$. Assume $\gamma_{-1} = 1$, then $\gamma_t = 1+\Lambda^2\eta_t^{2-(t+1)}$. By multiplying $\gamma_t$ on both sides of Eqn. 29, we have:

$$\gamma_t\eta_t\mathbb{E}\left[\left\|\nabla\mathcal{L}(\boldsymbol{\theta}^{t'})\right\|^2\right] \leq \gamma_{t-1}\mathbb{E}\left[\mathcal{L}(\boldsymbol{\theta}^t)-\mathcal{L}^*\right] \tag{30}$$
$$- \gamma_t\mathbb{E}\left[\mathcal{L}(\boldsymbol{\theta}^{t+1})-\mathcal{L}^*\right] + \gamma_t\Lambda^2\eta_t^2\sigma^2$$

Summing up the above equation from $t = 0, ..., T-1$, we have:

$$\sum_{t=0}^{T-1}\gamma_t\eta_t\mathbb{E}\left[\|\nabla\mathcal{L}(\boldsymbol{\theta}^{t'})\|^2\right] \leq \mathbb{E}\left[\mathcal{L}(\boldsymbol{\theta}^0)-\mathcal{L}^*\right] \tag{31}$$
$$+ \Lambda^2\sigma^2\sum_{t=0}^{T-1}\gamma_t\eta_t^2$$

Dividing both sides by $\sum_{t=0}^{T-1}\gamma_t\eta_t^2$, we have:

$$\min_{t=0,...,T-1}\mathbb{E}\left[\left\|\nabla\mathcal{L}(\boldsymbol{\theta}^{t'})\right\|^2\right] \tag{32}$$
$$\leq \frac{1}{\sum_{t=0}^{T-1}\gamma_t\eta_t}\sum_{t=0}^{T-1}\gamma_t\eta_t\left\|\nabla\mathcal{L}(\boldsymbol{\theta}^{t'})\right\|^2$$
$$\leq \frac{\mathbb{E}\left[\mathcal{L}(\boldsymbol{\theta}^0)-\mathcal{L}^*\right] + \Lambda^2\sigma^2\sum_{t=0}^{T-1}\gamma_t\eta_t^2}{\sum_{t=0}^{T-1}\gamma_t\eta_t}$$

Assume $\eta_t \equiv \eta$, then we have:

$$\sum_{t=0}^{T-1}\gamma_t\eta_t = \eta\sum_{t=0}^{T-1}(1+\Lambda^2\gamma_t^2)^{-(t+1)} \tag{33}$$
$$= \frac{\gamma}{1+\Lambda^2\eta^2}\frac{1-(1+\Lambda^2\eta^2)^{-T}}{1-(1+\Lambda^2\eta^2)^{-1}}$$
$$= \frac{1-(1+\Lambda^2\eta^2)^{-T}}{\Lambda^2\eta}$$

Note that $(1+\Lambda^2\eta^2)^{-T} \leq \frac{1}{2}$ and $\frac{x}{1+x} \leq \log(1+x)$, thus we have

$$\frac{\log(2)}{\log(1+\Lambda^2\eta^2)} \leq \frac{\log(2)(1+\Lambda^2\eta^2)}{\Lambda^2\eta^2} \leq T \tag{34}$$

From this, we can obtain:

$$\sum_{t=0}^{T-1}\gamma_t\eta_t \geq \frac{1}{2\Lambda^2\eta}, \text{ for } T \geq \frac{\log(2)(1+\Lambda^2\eta^2)}{\Lambda^2\eta^2} \tag{35}$$

Inserting the above equation into the Eqn. 32, we have:

$$\min_{t=0,...,T-1}\mathbb{E}\left[\left\|\nabla\mathcal{L}(\boldsymbol{\theta}^{t'})\right\|^2\right] \tag{36}$$
$$\leq 2\Lambda^2\eta\mathbb{E}\left[\mathcal{L}(\boldsymbol{\theta}^0)-\mathcal{L}*\right] + \eta\Lambda^2\sigma^2, \text{ for } T \geq \frac{\log(2)(1+\Lambda^2\eta^2)}{\Lambda^2\eta^2} \tag{37}$$

Setting $\eta = \frac{1}{\Lambda\sqrt{T-1}}$, we finally have:

$$\min_{t=0,\ldots,T-1} \mathbb{E}\left[\left\|\nabla\mathcal{L}(\boldsymbol{\theta}^{t'})\right\|^2\right] \tag{38}$$

$$\leq \frac{2\Lambda}{\sqrt{T-1}}\mathbb{E}\left[\mathcal{L}(\boldsymbol{\theta^0}) - \mathcal{L}_*\right] + \frac{\Lambda\sigma^2}{\sqrt{T-1}} \tag{39}$$

Thus, our method can readily reach to the Pareto Stationary point. $\qquad\square$

