# OpenReview forum: "Continual Optimization with Symmetry Teleportation for Multi-Task Learning"
_NeurIPS.cc/2025/Conference — NeurIPS 2025 poster_

### Official Review · Reviewer_WWrv · 2025-06-22

**Clarity:** 2
**Significance:** 2
**Originality:** 2
**Rating:** 4
**Confidence:** 3

**Summary:**

The paper 'Continual Optimization with Symmetry Teleportation for Multi-Task Learning' proposes to use symmetry teleportation [Zhao, 2022] to deal with optimization conflict and task imbalance, which are common problems in multi-task learning. As the authors point out, the original symmetry teleportation is not scalable to larger and general neural networks as it usually requires some invertibility assumptions. This paper proposes to implement symmetry teleportation on large scale neural networks using a LoRA method to circumvent some of the difficulties. They also developed historical trajectory reuse strategy (HTR) to mimic momentum-based optimizers like Adam. Some empirical experiments are presented to show the effectiveness of COST.

**Questions:**

Major:
1. In eq (4), how do you decide the RHS? How does the model performance and running time vary when you vary the bound on RHS?
2. eq (7) and and Fig 5 (b) seem inconsistent to me. Since $\mathcal{L}^{\star}_i $  is the loss before teleportation, shouldn't we have $\mathcal{L}^{\star}_i = \mathcal{L}_i$ before teleportation? Then shouldn't $L_t$ in Fig 5(b) starts from $(0,0)$? Please clarify.
3. eq (8) and eq (9) seem inconsistent to me. Why does $\mathcal{L}(\theta^*)$ disappear in eq (9)? It seems that you are suggesting $\max_{\varepsilon} |\mathcal {L}(\theta+\varepsilon) - \mathcal{L}(\theta)| = \max_{\varepsilon} |\mathcal {L}(\theta+\varepsilon)|$ which is not true.
4. In line 212, how do you choose $\tilde{n}$? Is there a systematic way to choose $\tilde{n}$ (maybe in terms of the dimension of $\theta$)? How does the model performance and running time vary when you vary $\tilde {n}$?
5. The results for experiments are averaged over 3 seeds, which seems not enough. Improvement in numerical experiments seems marginal when considering the error bar (e.g. in Table 1, $4.30 \pm 1.3$ vs $5.18$). Averaging over more seeds might reduce the error bar and give a clearer result.
6. minimizing $\mathcal{L}_t$ in eq (7) is only an approximation to the loss invariance assumption. What is the stopping criteria? In addition, how to compare the performance under this approximation with the performance under a true loss invariance assumption?
7. Is COST with HTR more sensitive to parameter tuning vs vanilla COST (lr, batch size, total iteration, etc)? Please clarify.


Minor:
1. Page 3 eq (2), $a = \mathrm{argmax}_{a\in A} \ldots$  is not a good notation. Consider using $a^*$ or something else on the left hand side. Similar issue in eq (6).
2. In Fig 5 (a), I cannot see the $K/2$ Condition bar for NYU v2. Is it 0? If so, please state it clearly.
3. line 197-199 is awkward, please correct the English. There are other typos and sentences that need to be fixed. Please proofread the paper carefully.
4. Please correct the format of Algorithm 1 in Appendix C.

**Ethical Concerns:**

["NO or VERY MINOR ethics concerns only"]

**Final Justification:**

The authors have addressed all my concerns with experiments. Thus I would like to increase my score.

**Limitations:**

Yes

**Quality:**

2

**Strengths And Weaknesses:**

Strengths:
1.The paper is well-organized and includes illustrations to clarify the ideas.
2. The idea to use symmetry teleportation to reconcile gradient conflicts and imbalance seems novel.


Weaknesses:
Overall, the quality of the paper can be greatly improved (English, notations). There are several major inconsistencies that need to be clarified. Current numerical experiments are insufficient to justify the significance of this paper. Several important unanswered questions are raised below.

---

> ### Author Rebuttal · Authors · 2025-07-31
>
> We sincerely thank Reviewer WWrv for the constructive and insightful comments. Below, we provide detailed point-by-point responses to each of your concerns.
>
> >* Q1: In eq (4), how do you decide the RHS? How does the model performance and running time vary when you vary the bound on RHS?
>
>   A1: In Figure 10 (Appendix), we provide an analysis of the trigger condition on the CelebA dataset. As shown, when the condition is varied from $K/2$, $K/3$, $K/4$, to $K/5$, the corresponding MTL performance scores are [0.93, 0.76, 0.82, 0.65], while the associated training time costs are [40.5, 51.7, 80.5, 112.1] minutes, respectively. Based on this trade-off between performance and computational efficiency, we adopt $K/2$ as the default trigger condition.
>
>
>
> >* Q2: Eq (7) and and Fig 5 (b) seem inconsistent to me. Since $L_i^\*$ is the loss before teleportation, shouldn't we have $L_i^\*=L_i$ before teleportation? Then shouldn't $L_t$  starts from (0,0)?
>
>   A2: We understand the reviewer's concern. Intuitively, if the full dataset or an identical batch order were used to compute both losses, we would expect $L_i^* = L_i$. However, in practice, a randomly shuffled batch-wise training paradigm is adopted, which leads to non-identical losses at the beginning due to inherent sampling variance.
>
>
>
> >* Q3: Eq (8) and eq (9) seem inconsistent to me. Why does $L(\theta)$ disappear in eq (9)? It seems that you are suggesting ....
>
>   A3: We apologize for the confusion — this was a typo introduced for ease of presentation. When applying the weighting factor $R_i$, the term $L(\theta)$ cannot be omitted. However, we would like to clarify that in our actual implementation, we do preserve the subtraction of $L(\theta)$. The relevant code snippet is provided below, and we are committed to releasing our code upon acceptance:
>
>   ```
>   sharp_loss.append(torch.stack([ratio_list[i] * torch.abs(lora_loss[i] - L0[i]) for i in range(n_tasks)]))
>   ```
>
>   Here, `ratio_list[i]` corresponds to $R_i$, `L0[i]` represents $L(\theta^\*)$, and `lora_loss[i]` is $L_i(\theta^\* + \epsilon)$.
>
>
>
> >* Q4: In line 212, how do you choose $\tilde{n}$? Is there a systematic way to choose $\tilde{n}$? How does the model performance and running time vary when you vary $\tilde{n}$?
>
>   A4: For operations akin to zero-order optimization, the gradient approximation is theoretically unbiased in expectation. However, when the number of samples is small (e.g., 1), the variance of the estimate becomes large, resulting in a potentially significant deviation from the true gradient. In our work, we set the number of samples based on common practices in related literature [1]. To further address the reviewer's concern, we conducted additional experiments with 10 and 15 samples. The results are presented below:
>
>   |             | mIoU  | Pix. Acc. | Abs. Err. | Rel. Err. | $\Delta m \\%$ |
>   | ----------- | ----- | --------- | --------- | --------- | ------------ |
>   | COST (n=5)  | 75.73 | 93.53     | 0.0133    | 31.53     | 4.30         |
>   | COST (n=10) | 73.86 | 93.14     | 0.0131    | 31.25     | 4.42         |
>   | COST (n=15) | 74.32 | 92.95     | 0.0125    | 32.52     | 4.19         |
>
>   As shown, the performance across $n=5$, $n=10$, and $n=15$ remains relatively stable, with the best results observed at $n=15$. However, a larger sample size also incurs greater computational overhead. Hence, we adopt $n=5$ as a trade-off between performance and efficiency.
>
>
>
> >* Q5: The results for experiments are averaged over 3 seeds, which seems not enough. Improvement in numerical experiments seems marginal when considering the error bar. Averaging over more seeds might reduce the error bar and give a clearer result.
>
>   A5: We would like to clarify that nearly all mainstream MTL approaches report results averaged over 3 random seeds (e.g., CAGrad, Nash-MTL, AlignMTL, FAMO, FairGrad, etc.). To further address the reviewer’s concern, we conducted and reported the following additional experiments:
>    (1) As a baseline, we faithfully re-ran FairGrad’s official code over three random seeds (consistent with both their implementation and ours), and reported the performance as *FairGrad-R* in the table below.
>    (2) As suggested, we further evaluated our method using two additional seeds, and reported the performance as *COST (5 seed)*.
>
>   |               | mIoU         | Pix. Acc.    | Abs. Err.       | Rel. Err.    | $\Delta m\\%$ |
>   | ------------- | ------------ | ------------ | --------------- | ------------ | ----------- |
>   | FairGrad*     | 75.72        | 93.68        | 0.0134          | 32.25        | 5.18        |
>   | FairGrad-R    | 75.14 ± 0.71 | 87.08 ± 0.11 | 0.0142 ± 0.0019 | 30.76  ± 1.60 | 5.58 ± 3.82 |
>   | COST (3 seed) | 75.71 ± 0.25 | 93.53 ± 0.12 | 0.0133 ± 0.0003 | 31.53 ± 1.85 | 4.30 ± 1.30 |
>   | COST (5 seed) | 75.12 ± 0.83 | 93.31 ± 0.31 | 0.0131 ± 0.0004 | 31.57 ± 1.53 | 4.32 ± 1.53 |
>
>   As shown, COST maintains stable and consistent performance when evaluated on more seeds, with only a slight increase in variance. Notably, under the same evaluation protocol, our method demonstrates substantially better stability compared to its baseline, *FairGrad-R*.
>
>
>
> >* Q6: minimizing $L_t$ in eq (7) is only an approximation to the loss invariance assumption. What is the stopping criteria? In addition, how to compare the performance under this approximation with the performance under a true loss invariance assumption?
>
>   A6: Since training is performed in a batch-wise manner, it is inherently difficult to determine whether the teleported point precisely lies at the symmetric position. Nevertheless, we would like to clarify that even the original symmetric teleportation work [2] does not achieve perfect loss invariance, primarily due to the inexact nature of gradient descent-based optimization.
>
>
>
> >* Q7: Is COST with HTR more sensitive to parameter tuning vs vanilla COST (lr, batch size, total iteration, etc)? Please clarify.
>
>   A7: To address the reviewer’s concern, we conducted the suggested comparison by varying batch sizes in the LoRA optimization and present the results below. As shown, HTR consistently enhances performance across different batch sizes. Furthermore, based on the observed standard deviations, COST with HTR exhibits greater robustness to batch size variation compared to COST without HTR.
>
>   |                | with HTR | w/o HTR  |
>   | -------------- | -------- | -------- |
>   | batch size = 4 | 4.68     | 4.92     |
>   | batch size= 8  | 4.30     | 4.65     |
>   | batch size= 16 | 4.97     | 5.62     |
>   | **std**        | **0.34** | **0.50** |
>
>
>
> >* Q8: In Fig 5 (a), I cannot see the K/ 2 Condition bar for NYU v2. Is it 0? If so, please state it clearly.
>
>   A8: Since the K/2 condition is only applied to CelebA and not to NYUv2, it is not displayed in the corresponding results. We will revise the text to clarify this distinction.
>
>
>
> >* Q9: Typos or inappropriate expression.
>
>   A9: Thank you for your reminder. We will correct these issues and carefully proofread the paper to ensure clarity and consistency.
>
>
>
>   If you have any further questions, please don’t hesitate to raise it.
>
>
>
>   **Reference**:
>
>   [1] ZO-AdaMM: Zeroth-Order Adaptive Momentum Method for Black-Box Optimization. NeurIPS 2018.
>
>   [2] Symmetry Teleportation for Accelerated Optimization. NeurIPS 2022.

---

> > ### Comment · Reviewer_WWrv · 2025-08-06
> >
> > Thank you for the responses. The authors have addressed most of my concerns.
> >
> > Q1: I would like to know how well can this K/2 be generalized to other tasks.
> >
> > Q6: what is the stopping criteria used in your experiments and how does varying that criteria change your overall running time and performance.

---

> > > ### Author Response · Authors · 2025-08-06
> > >
> > > Thank you for your reply. We would like to address your concerns as follows:
> > >
> > > > * Q1: I would like to know how well this K/2 condition generalizes to other tasks.
> > >
> > > A1: As mentioned in the original text, the K/2 condition is specifically adopted for CelebA to improve efficiency. For all other tasks, we use the *dominated conflict condition*, which is not triggered as frequently as on CelebA. Therefore, the K/2 condition is not required in those cases. We will clarify this distinction more explicitly in the revised version.
> > >
> > > > * Q6: What is the stopping criterion used in your experiments, and how does varying it affect overall running time and performance?
> > >
> > > A6: Following the original symmetric teleportation work [1], we adopt a fixed number of optimization steps during each teleportation phase. The rationale behind this design choice, as discussed in [1], is threefold:
> > >
> > > 1. **Efficiency**: It is computationally expensive to locate the exact loss-invariant point with the globally maximal gradient norm on a complex loss landscape.
> > > 2. **Practicality over optimality**: The goal of teleportation is to *improve* the current point, not necessarily to find the optimal point. A few optimization steps are often sufficient to increase the gradient norm and guide the training in a better direction.
> > > 3. **Empirical stability**: The original paper observed that the effectiveness of teleportation is not highly sensitive to the specific number of steps used.
> > >
> > > To further address your concern in the MTL setting, we are conducting an ablation study on this fixed-step criterion on the CityScapes. We will share the results in the coming days. Thank you for your patience.
> > >
> > >
> > >
> > > Reference:
> > >
> > > [1] Symmetry Teleportation for Accelerated Optimization. NeurIPS 2022.

---

> > > > ### Comment · Reviewer_WWrv · 2025-08-06
> > > >
> > > > Thanks for the reply. I appreciate the quick response and look forward to your results.

---

> > > > > ### Author Response · Authors · 2025-08-07
> > > > >
> > > > > Thank you for your patience. We now present the results on CityScapes in the table below. As observed, the MTL performance demonstrates a positive correlation with the number of teleportation steps (epoch), where a higher number of steps tends to yield better results. However, this improvement comes at the cost of increased computational overhead, potentially reducing overall efficiency. We plan to conduct further investigations by expanding the range of teleportation steps in future versions.
> > > > >
> > > > > | Fixed Teleportation Step (Epoch) | $\Delta m\\%$ |
> > > > > | ------------------- | ----------- |
> > > > > | 1                   | 4.74        |
> > > > > | 2 (default setting) | 4.30        |
> > > > > | 3                   | 3.96        |

---

> > > > > ### Author Response · Authors · 2025-08-08
> > > > >
> > > > > Dear Reviewer WWrv,
> > > > >
> > > > > As the deadline for the author–reviewer discussion phase approaches, we would like to kindly ask whether our responses have addressed your concerns.
> > > > >
> > > > > Best regards,
> > > > >
> > > > > Authors

---

### Official Review · Reviewer_sH9s · 2025-07-03

**Clarity:** 2
**Significance:** 3
**Originality:** 3
**Rating:** 4
**Confidence:** 3

**Summary:**

This paper proposes Continual Optimization with Symmetry Teleportation (COST), which seeks a new point on the loss-invariant contour to reduce gradient conflict. Besides, the authors proposes Historical Trajectory Reuse Strategy to keep the optimizer history. Experiments show improvements over previous methods.

**Questions:**

1. Why do we use L1 norm in eq. 7? What would happen if we use L2 norm?
2. What would happen if we relax the trigger to 0? By that I mean change the r.h.s. of eq. 4 to 0. Will it significantly improve the results?

**Ethical Concerns:**

["NO or VERY MINOR ethics concerns only"]

**Final Justification:**

After the rebuttal, I will keep my ratings unchanged. I encourage the authors to add the experiment results of Q&A5 in the next revision.

**Quality:**

2

**Strengths And Weaknesses:**

Strength:
1. The idea of transporting the parameters on the loss contour to resolve gradient conflict is reasonable and interesting. It seems a promising direction to push multi-task learning one step forward.
2. The authors did extensive experiments to compare COST with previous methods. This is very informative for readers.

Weaknesses:
1. Why do we even need LORA? We can simply optimize the backbone parameters with Eq.11 when conflict occurs.
2. In Section 5.2, I see no support on the statement 'These results might address another concern regarding COST, namely: Are the improvements brought about by COST rooted in the capability expansion facilitated by LoRA?' Could the authors elaborate that?
3. To prove that the improvement is not from LORA, the authors should add the following ablation study: when conflict occurs, instead of optimize eq. 11 with lora, directly optimize the original objective eq. 5 with lora, and then compare the difference in performance.

---

> ### Author Rebuttal · Authors · 2025-07-31
>
> We sincerely thank Reviewer sH9s for the constructive and insightful comments. Below, we provide detailed point-by-point responses to each of your concerns.
>
> >* Q1: Why do we even need LORA? We can simply optimize the backbone parameters with Eq.11 when conflict occurs.
>
>   A1: Yes, it is feasible to directly optimize the backbone for symmetric teleportation. In our work, we employ LoRA primarily for efficiency purposes.
>
>
>
> >* Q2&Q3: In Section 5.2, I see no support on the statement 'These results might address ... by LoRA?' Could the authors elaborate that? To prove that the improvement is not from LORA, the authors should add the following ablation study: ...
>
>   A2&A3: Thank you for your valuable suggestion. We have conducted the comparison experiments accordingly, and the results are presented below:
>
>   |                | mIoU  | Pix. Acc. | Abs. Err. | Rel. Err. | $\Delta m \\%$ |
>   | -------------- | ----- | --------- | --------- | --------- | ------------ |
>   | COST (Eqn. 5)  | 75.73 | 93.53     | 0.0133    | 31.53     | 4.30         |
>   | LoRA (Eqn. 11) | 75.70 | 93.66     | 0.0134    | 32.61     | 5.40         |
>
>   As shown, applying LoRA alone does not lead to performance gains. One possible explanation is that the paradigm of alternating between pre-training and LoRA-based fine-tuning may not be optimal for this setting.
>
>
>
> >* Q4: Why do we use L1 norm in eq. 7? What would happen if we use L2 norm?
>
>   A4: Thank you for your valuable suggestion. We conducted an additional experiment by replacing the L1 norm in Eq. (7) with the L2 norm, and present the results below. As shown, the L2 norm performs slightly worse than the L1 norm, but still achieves SOTA performance.
>
>   |                | mIoU  | Pix. Acc. | Abs. Err. | Rel. Err. | $\Delta m \\%$ |
>   | -------------- | ----- | --------- | --------- | --------- | ------------ |
>   | COST (L1 norm) | 75.73 | 93.53     | 0.0133    | 31.53     | 4.30         |
>   | COST (L2 norm) | 74.43 | 92.83     | 0.0128    | 32.13     | 4.46         |
>
>
>
> >* Q5: What would happen if we relax the trigger to 0? By that I mean change the r.h.s. of eq. 4 to 0. Will it significantly improve the results?
>
>   A5：As illustrated in Figure 5, setting the right-hand side (RHS) of Eq. (4) to zero results in frequent teleportations, with a conflict ratio of 97% per epoch. Due to the limited rebuttal time, we were unable to conduct further experiments under this configuration. Nevertheless, as shown in Figure 10 (Appendix), reducing the threshold from $K/2$ to $K/3$, $K/4$, and $K/5$ yields results of 0.76, 0.82, and 0.65, respectively. These outcomes are generally consistent with our expectations and further support the rationale behind our design choices.
>
>
>
>   If you have any further questions, please don’t hesitate to raise it.

---

### Official Review · Reviewer_G25M · 2025-07-03

**Clarity:** 2
**Significance:** 3
**Originality:** 2
**Rating:** 4
**Confidence:** 4

**Summary:**

To handle the important issues of optimization conflict in MTL tasks, this paper proposes a plug-and-play solution COST, the key point of which is applying symmetry teleportation. The method employs LoRA to propose a practical and effective solution for symmetry teleportation, and adopts historical trajectory reuse strategy to keep the advantages of optimizers like Adam. Several experiments indicate the performance of COST.

**Questions:**

1、How can the design of R in Equ.10 mitigate imbalance issues and what’s the performance of this design?
2、Does the delayed start strategy provide any beneficial effects on model effectiveness or convergence, in addition to saving training time?
3、What’s the performance of COST-weak when relaxing from [K/2, K/3, K/4, K/5]？Does COST-dominated offer the best performance in most datasets?

**Ethical Concerns:**

["NO or VERY MINOR ethics concerns only"]

**Final Justification:**

The authors provided evidence that the design of R effectively alleviates imbalance issues in multi-task learning scenarios and achieves state-of-the-art performance in overall metrics. However, COST shows a poorer performance on Segmentation task in NYUv2 (Table 2), indicating the seesaw phenomenon between different tasks. Considering the concerns on task-specific robustness and training efficiency, I maintain my original rating.

**Limitations:**

Yes.

**Paper Formatting Concerns:**

None.

**Quality:**

3

**Strengths And Weaknesses:**

Strengths
1、This work mitigates optimization conflicts in MTL by applying symmetry teleportation and reformulating it through LoRA to obtain a practical implementation. Besides, the advantages of optimizers like Adam can be retained by the proposed historical trajectory reuse strategy.
2、This method is evaluated with several mainstream MTL benchmarks and compared to a set of optimization methods, and performs competitively.
3、This paper is well written with detailed description, proofs and visualization.

Weaknesses
1、It seems that the paper does not address the issue of task imbalance in MTL. While the model achieves good performance in most overall metrics, the model's performance varies significantly across different tasks, which raises concerns about the robustness and stability of the proposed approach.
2、This method is time-consuming, especially when the trigger condition relaxes. Even with delayed start strategy and frequency control strategy, COST
introduces an additional 30% of the training time, making it hard to use in large-scale datasets.

---

> ### Author Rebuttal · Authors · 2025-07-31
>
> We sincerely thank Reviewer G25M for the constructive and insightful comments. Below, we provide detailed point-by-point responses to each of your concerns.
>
> >* Q1: It seems that the paper does not address the issue of task imbalance in MTL. While the model achieves good performance in most overall metrics, the model's performance varies significantly across different tasks, which raises concerns about the robustness and stability of the proposed approach.
>
>   A1: We would like to clarify that the designs in Equation (9) and Equation (10) are specifically tailored to address task imbalance. To facilitate balanced gradient exploration, we first compute the re-weighting factor $R$ based on task gradient norms (Eqn. (10)) prior to teleportation. This factor is then used to re-weight the task sharpness in Eqn. (9), guiding the search toward the most balanced descent direction during teleportation. Additionally, we emphasize that COST consistently achieves state-of-the-art performance across all evaluated datasets, with notable improvements on CityScapes, NYUv2, and CelebA.
>
>
>
> >* Q2: This method is time-consuming, especially when the trigger condition relaxes. Even with delayed start strategy and frequency control strategy, COST introduces an additional 30% of the training time, making it hard to use in large-scale datasets.
>
>   A2: We acknowledge that our method incurs an approximately 30% increase in training time, which we recognize as a limitation and have explicitly discussed in the paper. However, we believe this additional cost is justified by the substantial benefits it offers: (1) Our approach explores a **novel perspective in MTL**, fundamentally distinct from existing optimization-based MTL methods, whose effectiveness remains a subject of debate [1,2]. (2) By incorporating and adapting symmetric teleportation into deep MTL, our method not only consistently achieves state-of-the-art performance across widely adopted MTL benchmarks, but also remains orthogonal and complementary to existing optimization-based approaches.
>
>
>
> >* Q3: How can the design of R in Equ.10 mitigate imbalance issues and what’s the performance of this design?
>
>   A3: Please refer to **A1** for further clarification. As additionally suggested, we conducted an ablation study to assess the impact of the re-weighting factor $R$. The results are presented below:
>
>   |              | mIoU  | Pix. Acc. | Abs. Err. | Rel. Err. | $\Delta m \\%$ |
>   | ------------ | ----- | --------- | --------- | --------- | ------------ |
>   | COST         | 75.73 | 93.53     | 0.0133    | 31.53     | 4.30         |
>   | COST w/o $R$ | 74.53 | 93.04     | 0.0137    | 33.78     | 7.58         |
>
>
>
>   As shown, removing the re-balancing design leads to a notable performance degradation, indicating the effectiveness of our proposed re-weighting mechanism.
>
>
>
> >* Q4: Does the delayed start strategy provide any beneficial effects on model effectiveness or convergence, in addition to saving training time?
>
>   A4: Currently, the delayed start strategy is adopted purely as a time-saving mechanism.
>
>
>
> >* Q5: What’s the performance of COST-weak when relaxing from [K/2, K/3, K/4, K/5]？Does COST-dominated offer the best performance in most datasets?
>
>   A5：We would like to clarify that we did not evaluate COST-weak on multiple datasets for the following reasons:
>    (1) Defining weak conflicts becomes increasingly ambiguous and challenging in scenarios involving more than two tasks.
>    (2) Even under the dominated conflict condition, CelebA already exhibits frequent teleportation triggering, incurring considerable computational overhead—this would be exacerbated under the weak condition.
>
>   Given the limited rebuttal timeframe, it is infeasible for us to conduct such extensive additional experiments. Moreover, as illustrated in **Figure 2**, weak conflicts may not pose a significant issue when task gradients exhibit approximately equal norms. Finally, **Table 7** already presents a comparison between COST-dominated and COST-weak, showing that the former substantially outperforms the latter.
>
>
>
>   If you have any further questions, please don’t hesitate to raise it.
>
>
>
>   **Reference**:
>
>   [1] Do Current Multi-Task Optimization Methods in Deep Learning Even Help? NeurIPS 2022.
>
>   [2] Can Optimization Trajectories Explain Multi-Task Transfer? TMLR 2024.

---

> > ### Comment · Reviewer_G25M · 2025-08-06
> >
> > Thanks for the detailed responses. I will keep my positive score.

---

> > > ### Author Response · Authors · 2025-08-06
> > >
> > > Thank you for your positive feedback!

---

### Official Review · Reviewer_Zv8Z · 2025-07-22

**Clarity:** 3
**Significance:** 3
**Originality:** 3
**Rating:** 5
**Confidence:** 3

**Summary:**

The paper proposes COST (Continual Optimization with Symmetry Teleportation) to tackle gradient conflict and task imbalance in multi-task learning (MTL). When conflicts arise, the model is “teleported” to an alternative point on the same loss level set (i.e., task losses unchanged) with better gradient conditions. Teleportation is implemented using LoRA modules, optimizing a loss-invariance and gradient-maximization objective. The authors introduce a historical trajectory reuse (HTR) strategy to preserve optimizer momentum (e.g., for Adam). COST is a plug-and-play framework tested on several datasets (CityScapes, NYUv2, CelebA, QM9), showing consistent improvements over strong MTL baselines.

**Questions:**

1. Please quantify how often teleportation is triggered in practice and its cumulative time cost. Could more dynamic or adaptive frequency strategies be explored?

2. Can the authors better explain the $L_g$ surrogate used to encourage stronger gradients after teleportation. What metric is being maximized in practice?

3. How sensitive is performance to LoRA rank? Why did LoRA outperform LoHa and OFT? Would dynamic rank tuning help?

4. How might COST be adapted to models where LoRA/PEFT is less applicable (e.g., GNNs or Transformers)?

5. Could imperfect teleportation or over-teleportation lead to optimization instability? Any examples observed?

**Ethical Concerns:**

["NO or VERY MINOR ethics concerns only"]

**Final Justification:**

I have read the author response and have no further questions. I'll maintain my score.

**Limitations:**

Yes

**Quality:**

3

**Strengths And Weaknesses:**

Quality:
- The paper shows Strong theoretical motivation: targets dominated conflict with formal definitions.
- The method enjoys practical and scalable implementation using LoRA.
- The authors perform extensive experiments on vision and graph tasks with solid baselines.
- The paper performs ablations and auxiliary studies (e.g., trigger frequency, PEFT alternatives) enhance rigor.

Limitation: COST introduces 30% training time overhead and added complexity.

Significance
- The paper addresses a central MTL challenge in a novel way.
- Plug-and-play nature means broad applicability across existing MTL optimizers.
- The authors show strong empirical results on datasets with both few and many tasks.
- The method is practical for many MTL settings, though overhead may limit adoption in large-scale training.

Originality
- The paper is the first to propose a practical symmetry teleportation method for deep MTL.

---

> ### Author Rebuttal · Authors · 2025-07-31
>
> We sincerely thank Reviewer Zv8Z for the constructive and insightful comments. Below, we provide detailed point-by-point responses to each of your concerns.
>
> >* Q1: Please quantify how often teleportation is triggered in practice and its cumulative time cost. Could more dynamic or adaptive frequency strategies be explored?
>
>   A1: As shown in Figure 5, teleportation is triggered infrequently when using the dominated conflict condition on NYUv2 and the $K/2$ condition on CelebA. We further report the per-epoch computational overhead of our method on CelebA in Figure 10 (Appendix), which confirms its efficiency. While our current design uses fixed conflict-triggering criteria, recent studies [1,2] suggest that it may not be necessary to apply MTL at every optimization step. In the same spirit, not every conflicting step may require teleportation. Designing dynamic triggering strategies could therefore be beneficial, though this remains an under-explored area. We thank the reviewer for this insightful suggestion and plan to investigate it further in future work.
>
>
>
> >* Q2: Can the authors better explain the $L_g$ surrogate used to encourage stronger gradients after teleportation. What metric is being maximized in practice?
>
>   A2: In our symmetric teleportation framework, we optimize two objectives: loss invariance $L_t$ and balanced gradient maximization $L_g$. For the design of $L_g$, we first compute a re-weighting factor $R$ based on task gradient norms (Eqn. (10)) prior to teleportation. This factor is then used to re-weight task sharpness in Eqn. (9), thereby guiding the search toward a direction that maximizes balanced gradients during teleportation.
>
>   Given that the negative gradient direction represents the steepest local descent, we use sharpness as a surrogate to approximate this descent direction. Specifically, sharpness is estimated by sampling perturbations from the unit sphere multiple times and taking the maximum perturbed loss as the sharpness measure. This approach shares a similar motivation with point-wise gradient estimators commonly used in online optimization literature [3,4].
>
>
>
> >* Q3: How sensitive is performance to LoRA rank? Why did LoRA outperform LoHa and OFT? Would dynamic rank tuning help?
>
>   A3: To address the reviewer’s concern, we conducted a comparison of LoRA ranks and present the results below:
>
>   |             | mIoU  | Pix. Acc. | Abs. Err. | Rel. Err. | $ \Delta m \\% $ |
>   | ----------- | ----- | --------- | --------- | --------- | ------------ |
>   | COST (R=5)  | 75.73 | 93.53     | 0.0133    | 31.53     | 4.30         |
>   | COST (R=10) | 73.67 | 92.87     | 0.0135    | 30.00     | 4.16         |
>   | COST (R=15) | 73.96 | 92.92     | 0.0134    | 30.13     | 4.04         |
>
>   As observed, increasing the rank leads to marginal improvements in MTL performance; however, it also results in increased training cost. Therefore, we adopt the default setting (R=5) to maintain a balance between performance and efficiency.
>
>   While a deeper investigation into LoRA is beyond the scope of this work, we note that similar findings have been reported in recent PEFT studies [5,6], where LoRA consistently ranks among the top-performing approaches. Intuitively, dynamically adjusting the rank may be beneficial, as the difficulty of finding a desirable symmetric point during teleportation may vary across steps. We thank the reviewer for this insightful suggestion and will consider exploring this direction in future work.
>
>
>
> >* Q4: How might COST be adapted to models where LoRA/PEFT is less applicable (e.g., GNNs or Transformers)?
>
>   A4: We would like to clarify that LoRA can indeed be applied to Transformer architectures. We understand and appreciate the reviewer’s concern regarding its applicability in graph-based models. Currently, most GNN frameworks offer limited support for LoRA and other PEFT techniques. To accommodate this, in our design, we apply LoRA only to the final linear head of the GNN.
>
>   For instance, in one of our evaluated datasets, **QM9**, which employs a GNN for regression tasks, we restrict teleportation and LoRA adaptation to the last linear layer. Despite this minimal intervention, our approach still achieves notable performance improvements, suggesting the effectiveness of even partial LoRA integration in non-Transformer models.
>
>
>
> >* Q5: Could imperfect teleportation or over-teleportation lead to optimization instability? Any examples observed?
>
>   A5: In our current experimental setting, we did not observe optimization instability. However, to proactively address the reviewer’s concern, we manually constructed an imperfect teleportation scenario. Specifically, we performed early stopping halfway through the LoRA optimization phase, and present the resulting performance comparison below:
>
>   |                   | mIoU  | Pix. Acc. | Abs. Err. | Rel. Err. | $\Delta m \\%$ |
>   | ----------------- | ----- | --------- | --------- | --------- | ------------ |
>   | COST              | 75.73 | 93.53     | 0.0133    | 31.53     | 4.30         |
>   | COST (Early Stop) | 74.40 | 92.96     | 0.0134    | 32.54     | 6.04         |
>
>   As shown, **imperfect teleportation—caused by premature termination of LoRA optimization—results in degraded MTL performance**. This observation supports the importance of fully optimizing teleportation steps to ensure the effectiveness of our method.
>
>
>
> >* Q6: Limitation: COST introduces 30% training time overhead and added complexity.
>
>   A6: We acknowledge that our method incurs an approximately 30% increase in training time, which we recognize as a limitation and have explicitly discussed in the paper. However, we believe this additional cost is justified by the substantial benefits it offers: (1) Our approach explores a **novel perspective in MTL**, fundamentally distinct from existing optimization-based MTL methods, whose effectiveness remains a subject of debate [7,8]. (2) By incorporating and adapting symmetric teleportation into deep MTL, our method not only consistently achieves state-of-the-art performance across widely adopted MTL benchmarks, but also remains orthogonal and complementary to existing optimization-based approaches.
>
>
>
>   If you have any further questions, please don’t hesitate to raise it.
>
>
>
>   **Reference**:
>
>   [1] Multi-Task Learning as a Bargaining Game. ICML 2022.
>
>   [2] Injecting Imbalance Sensitivity for Multi-Task Learning. arXiv:2503.08006, 2025.
>
>   [3]  Introduction to Online Convex Optimization. Foundations and Trends® in Optimization, 2016.
>
>   [4] Online convex optimization in the bandit setting: gradient descent without a gradient. arXiv preprint cs/0408007, 2004.
>
>   [5] Empirical analysis of the strengths and weaknesses of peft techniques for llms. arXiv:2304.14999, 2023.
>
>   [6] Quantum-PEFT: Ultra parameter-efficient fine-tuning. ICLR 2025.
>
>   [7] Do Current Multi-Task Optimization Methods in Deep Learning Even Help? NeurIPS 2022.
>
>   [8] Can Optimization Trajectories Explain Multi-Task Transfer? TMLR 2024.

---

> ### Author Response · Authors · 2025-08-08
>
> Dear Reviewer Zv8Z,
>
> As the deadline for the author–reviewer discussion phase approaches, we would like to kindly ask whether our responses have addressed your concerns.
>
> Best regards,
>
> The Authors of Paper 9185

---

### Author Response · Authors · 2025-08-09

Dear AC and Reviewers,

As we reach the conclusion of the discussion phase, we wish to convey our genuine gratitude to all reviewers for your thoughtful feedback and the generous time you devoted despite demanding schedules. We are pleased that the concerns you raised have been resolved. We would also like to warmly thank the AC for your active role in guiding constructive exchanges. Your engagement is instrumental in nurturing the continued advancement of our research community.

Best regards,

The Authors of Paper 9185

---

### Decision · Program_Chairs · 2025-09-17

**Decision:**

Accept (poster)

**Comment:**

This paper proposes COST, a continual optimization framework using symmetry teleportation to mitigate gradient conflict and task imbalance in multi-task learning. Reviewers highlighted the method’s novelty, strong empirical performance across multiple benchmarks, and its plug-and-play nature. Concerns were raised about training overhead, task imbalance, and ablation coverage, but the authors provided detailed responses and additional experiments that addressed most of these points. During the rebuttal phase, all reviewers actively engaged in the discussion phase, and several updated their scores positively after rebuttal. Overall, the contribution is solid, with clear motivation and extensive experiments. The AC encourages the authors to incorporate reviewer feedback and clarifications into the final version.